**Biogeochemical evidence of anaerobic methane oxidation on active submarine mud**
**volcanoes on the continental slope of the Canadian Beaufort Sea**
Dong-Hun Lee[1], Jung-Hyun Kim[2,*], Yung Mi Lee[2], Alina Stadnitskaia[3], Young Keun Jin[2],
Helge Niemann[4,5], Young-Gyun Kim[6], Kyung-Hoon Shin[1,**]
[1]Hanyang University, 425-791 Ansan, South Korea
[2]KOPRI Korea Polar Research Institute, 406-840 Incheon, South Korea
[3]European Council for Maritime Applied R&D, Rue Marie de Bourgogne 52-54, B-1000
Bruxelles, Belgium
[4]NIOZ Royal Netherlands Institute for Sea Research, Department of Marine Microbiology
and Biogeochemistry, NL1790 AB Den Burg, Texel, The Netherlands
[5]CAGE - Centre for Arctic Gas Hydrate, Environment and Climate, Department of Geology,
UiT The Arctic University of Norway, 9037 Tromsø, Norway
[6]Research Institute for Earth Resources, Kangwon National University, 24341, Chuncheon,
South Korea
[*]Corresponding author:
Tel.: +82 32 760 5377
jhkim123@kopri.re.kr
[**]Corresponding author:
Tel.: +82 31 400 5536
shinkh@hanyang.ac.kr

**Abstract**

In this study, we report lipid biomarker patterns and phylogenetic identities of key microbial communities mediating anaerobic oxidation of methane (AOM) in active mud volcanoes (MVs) on the continental slope of the Canadian Beaufort Sea. A steep depletion in sulfate concentrations within 0.7 m in core depths and highly negative carbon isotopic compositions ($\delta^{13}$C) of *sn*-2- and *sn*-3-hydroxyarchaeol ($-114$ ‰ to $-82$ ‰) compared with those of methane ($-64$ ‰) indicated the presence of methanotrophic archaea involved in sulfate dependent–AOM, albeit in a small amount. The ratio of *sn*-2-hydroxyarchaeol to archaeol ($>1$) and operational taxonomic units (OTUs) indicated that archaea of the ANME-2c and ANME-3 clades were involved in AOM. Higher $\delta^{13}$C values of archaeol and biphytanes (BPs) ($-55.2 \pm 10.0$ ‰ and $-39.3 \pm 13.0$ ‰, respectively) suggested that archaeal communities were also assimilating AOM-derived inorganic carbon. Furthermore, the distinct distribution patterns of methanotrophs in the three MVs appears to be associated with varying intensities of ascending gas fluids. Further studies are needed to investigate the diversity and distribution of the AOM-related communities in detail and to clarify their preferred habitats in the uppermost surface sediments of Beaufort Sea MV systems.

**Keywords:** Arctic, Beaufort Sea, submarine mud volcano, methane, anaerobic oxidation of methane (AOM), lipid biomarkers, 16S rRNA

## 1 Introduction

Mud volcanoes (MVs) are kilometer-scale, low-temperature, seepage-related geomorphological features that provide some of the most remarkable indications of fluid venting (Ivanov et al., 1998). They are similar to magmatic volcanoes - their eruptions are powerful - but instead of lava, mud volcanoes expel a complex mixture of products, including hydrocarbon gases (e.g. methane and wet hydrocarbons), hydrogen sulfide, carbon dioxide, petroleum, pore water, and mud. The roots of MVs can reach depths of up to 20 km (Shnukov et al., 2005); thus they provide key information about the geological history of the area and its possible hydrocarbon potential (Ivanov et al.,1992, 1998). Comprehensive investigations of numerous on- and off-shore MV provinces have revealed the overwhelming input of hydrocarbon gases in their formation. Eruptions often manifest as a catastrophic emission of fluids consisting of hydrocarbon gases (especially methane), hydrogen sulfide, carbon dioxide, petroleum products, water, and a complex mixture of sediments, so-called "mud breccia" (Akhmanov,1996; Akhmanov and Woodside, 1998; Ivanov et al.,1998). The occurrence of active MVs could constitute a significant portion of the geological sources of global atmospheric methane emissions (Kopf, 2002; Milkov et al., 2003). In the Arctic Ocean, where the temperature of the bottom water has been increasing (Levitus et al., 2000; Steele et al., 2008; Westbrook et al., 2009; Polyakov et al., 2010; Thatcher et al., 2013, Somavilla 2013), concern has been raised that the warming water will cause the disintegration of sediment-bound methane gas hydrates (Marín-Moreno et al., 2016). That would lead to higher methane concentrations/fluxes in surface sediments and thus the ascending methane would quickly be released into the water column and potentially the atmosphere (Niemann et al., 2006; Felden et al., 2010). The submarine MVs is therefore of considerable interest in global warming scenarios, since methane is a greenhouse gas that is >20 times more potent than carbon dioxide (Wuebbles and Hayhoe, 2002; Etminan et al., 2016). Accordingly, MV

sediments can be regarded as a model system for studying the biogeochemical dynamics of sediments characterized by high methane fluxes.

Across the Canadian Beaufort continental slope, active MVs were discovered at water depths of ~282 m, ~420 m, and ~740 m during the multibeam bathymetric mapping surveys conducted in 2009 and 2010 (Campbell et al., 2009). They were named with respect to their water depths, i.e., MV282, MV420, and MV740 (Blasco et al., 2013; Saint-Ange et al., 2014). Previous investigations based on sediment coring and mapping with an autonomous underwater vehicle (AUV) and a remotely operated vehicle (ROV) showed that these MVs are young and active edifices characterized by ongoing eruptions (Paull et al., 2015). The gas ascending via these MVs consists of >95 % methane with $\delta^{13}C_{CH4}$ values of −64 ‰ (Paull et al., 2015), indicating a microbial methane source (Whiticar, 1999). Siboglinid tubeworms and white bacteria mats were reported at MV420 (Paull et al., 2015). Those organisms typically consume sulfide and are thus often associated with elevated anaerobic methanotrophy in near-surface sediments because sulfide is one end product of the anaerobic oxidation of methane (AOM), with sulphate as the terminal electron acceptor (Boetius and Wenzhöfer, 2013; Paull et al., 2015). AOM is mediated by several clades of anaerobic methanotrophic archaea (ANME) that typically form syntrophic associations with sulphate-reducing partner bacteria (Knittel and Boetius, 2009):

$$CH_4 + SO_4^{2-} \rightarrow HCO_3^- + HS^- + H_2O$$

A powerful tool to investigate AOM communities in sediments is the analysis of membrane lipids combined with their compound-specific carbon isotopic composition ($\delta^{13}C$), which can be used to chemotaxonomically infer community composition (Niemann and Elvert, 2008 and references therein). In particular, low $\delta^{13}C$ values in AOM-derived lipids are

widely used to trace AOM in ancient (e.g. Zhang et al., 2003; Stadnitskaia et al.,2008a,b;
Himmler et al., 2015) and modern seep settings (e.g. Hinrichs and Boetius, 2002; Niemann et
al., 2005; Chevalier et al., 2011, 2014). Although the ebullition of methane from the Beaufort
Sea MVs has been documented before (Paull et al., 2015), the sediment methane dynamics,
including the role of AOM as a barrier against uprising methane in these systems, has not
been investigated.

In this study, we thus investigated three sediment cores recovered from active MVs on

the continental slope of the Canadian Beaufort Sea during the ARA05C expedition with the
R/V ARAON in 2014. By conducting integrated lipid and nucleic acid analyses with bulk
geochemical parameters, our study sheds light on the specific archaeal communities involved
in AOM at active MVs in the Canadian Beaufort Sea.

**2 Material and Methods**
2.1 Sample collection

Three sediment cores were recovered using a gravity corer during the ARA05C

expedition of the South Korean icebreaker R/V ARAON in the Canadian Beaufort Sea in
August 2014 (Fig. 1A-C). Core ARA05C-10-GC (70°38.992'N, 135°56.811'W, 282 m water
depth, 221 cm core length), core ARA05C-01-GC (70°47.342'N, 135°33.952'W, 420 m water
depth, 272 cm core length), and core ARA05C-18-GC (70°48.082'N, 136°05.932'W, 740 m
water depth, 300 cm core length) were retrieved from the active MV sites MV282, MV420
and MV740, respectively. Upon recovery, all sediment cores showed active degassing (Fig.
1D). When the sediment cores were split, we observed a mousse-like texture in cores
ARA05C-10-GC and ARA05C-01-GC, related to outgassing as a result of the pressure
change during recovery. Gas hydrates in the shape of about ≤2 cm thick isolated veins were
observed at the bottom (230 to 300 cm) of core ARA05C-18-GC. The split sediment cores
were lithologically described, and then subsampled for total organic carbon (TOC), lipid
biomarkers and 16S rRNA gene sequences on board. After subsampling, sediment samples
were stored at $-20°C$ for geochemical analyses and at $-80°C$ for microbial analyses.

2.2 Bulk geochemical analysis
Sediment samples were freeze-dried and homogenized using an agate mortar prior to
the TOC analyses. Sediment samples (~1 g) were then treated with 8 mL 1N HCl to remove
carbonates before measuring the TOC content and its isotopic composition using an
elemental analyzer (EuroEA3028, Eurovector, Milan, Italy) connected to an isotope ratio
mass spectrometer (Isoprime, GV Instruments, Manchester, UK). All isotope ratios of TOC
are reported using the δ-notation (per mil) with respect to the Vienna Pee Dee Belemnite
(VPDB). The analytical errors (standard deviations of repeated measurements of the internal
standard IAEA CH$_6$) were smaller than $\pm0.1$ wt.% for TOC, and $\pm0.1$ ‰ for $\delta^{13}C_{TOC}$.

2.3 Lipid extraction and purification
The homogenized sediment samples (ca. 10 g) were extracted with an accelerated solvent
extractor (Dionex ASE 200, Dionex Corporation, Sunnyvale, CA) using a solvent mixture of
9:1 (v:v) dichloromethane (DCM) to methanol (MeOH) at a temperature of 100°C and a
pressure of $7.6\times10^6$ Pa. The total lipid extract was dried over anhydrous Na$_2$SO$_4$ and was
treated with tetrabutylammonium sulfite reagent to remove elemental sulfur. An aliquot was
chromatographically separated into apolar and polar fractions over an Al$_2$O$_3$ (activated for 2 h
at 150°C) column with solvents of increasing polarity. The apolar fraction was eluted using
hexane:DCM (9:1, v:v), and the polar fraction was recovered with DCM:MeOH (1:1, v:v) as
eluent. After column separation, 40 µl of 5α-androstane (10 µg/mL) were added to the apolar
fraction as an internal standard. The polar fraction was divided into two aliquots, to which
either $C_{22}$ 7,16-diol (10 µg/mL) or $C_{46}$ GDGT (10 µg/mL) was added as an internal standard.
Half of the polar fraction containing $C_{22}$ 7,16-diol was dried and silylated with 25 µL N,O-
bis(trimethylsilyl)trifluoroacetamide (BSTFA) and 25 µL pyridine before heating it to 60℃
for 20 min to form trimethylsilyl derivatives. The second half of the polar fraction containing
$C_{46}$ GDGT was re-dissolved by sonication (5 min) in hexane:isopropanol (99:1, v:v) and then
filtered with a 0.45-µm PTFE filter. Afterwards, an aliquot of the filtered fraction was treated
with HI following the procedure described by Kaneko et al. (2011) in order to cleave ether
bonds from glycerol dialkyl glycerol tetraethers (GDGTs), thereby releasing biphytanes (BPs)
which can be analyzed by a gas chromatography (GC).

2.4 Identification and quantification of lipid biomarkers
All apolar and polar fractions were analyzed using a Shimazu GC (Shimazu Corporation,
Kyoto, Japan) equipped with a splitless injector and a flame ionization detector for compound
quantification. A fused silica capillary column (CP-sil 5 CB, 25-m length, 0.32-mm i.d., and
0.12-µm film thickness) was used with He (1.3 mL/min) as a carrier gas. The samples were
injected under constant flow at an initial oven temperature of 70℃. The GC oven
temperature was subsequently raised to 130℃ at a rate of 20℃/min, and then to 320℃ at
4℃/min with a final hold time of 15 min. Concentrations were obtained by comparing the
peak area of each compound with that of 5α-androstane for the apolar fraction and $C_{22}$ 7,16-
diol for the polar fraction. Compound identifications for the apolar, silylated and BP polar
fractions were conducted using a Shimazu GC connected to a GCMS-QP2010 mass
spectrometer (MS) operated at 70 eV (cycle time of 0.9 s, resolution of 1000) with a mass
range of $m/z$ 50–800. The samples were subjected to the same temperature conditions and
capillary column described for GC analysis. Molecular structures were determined by

comparing their mass spectral fragmentation patterns and retention times with previously published data.

An aliquot of the filtered polar fractions was analyzed by high-performance liquid chromatography–atmospheric pressure positive ion chemical ionization–mass spectrometry using an Agilent 6120 Series LC/MSD SL system (Agilent Technologies, Santa Clara, CA) equipped with an auto-injector and Chemstation chromatography manager software. Separation was achieved on two UHPLC silica columns (2.1 × 150 mm, 1.7 μm), fitted with 2.1 × 5 mm pre-columns of the same material and maintained at 30℃. Injection volumes varied from 1 μL. GDGTs were eluted isocratically with 82% A and 18% B for 25 min, followed by a linear gradient to 35% B over 25 min, then to 100% B over 30 min, and finally maintained for 20 min, where A = hexane and B = hexane:2-propanol (90:10, v:v). The flow rate was 0.2 mL/min, with a total run time of 90 min. After each analysis, the column was cleaned by back-flushing hexane:2-propanol (90:10, v:v) at 0.2 mL/min for 20 min. Conditions for APCI-MS were as follows: nebulizer pressure 60 psi, vaporizer temperature 400℃, drying gas ($N_2$) flow 6 mL/min and temperature 200℃, capillary voltage −3.5 kV, corona 5 μA (~3.2 kV). Detection was achieved in single ion monitoring of $[M + H]^+$ ions (dwell time 35 ms), as described by Schouten et al. (2007). GDGTs were quantified by integrating peak areas and using the internal standard according to Huguet et al. (2006).

2.5 Compound-specific stable carbon isotope analysis

The $\delta^{13}C$ values of selected compounds were determined by GC/combustion/isotope ratio mass spectrometry (GC-C-IRMS), as described by Kim et al. (2017). An IRMS (Isoprime, GV Instruments, UK) was connected with a GC (Hewlett Packard 6890 N series, Agilent Technologies, Santa Clara, CA) via a combustion interface (glass tube packed with copper oxide (CuO), operated at 850℃). The samples were subjected to the same

temperature conditions and capillary column described for the GC and GC-MS analyses.
Calibration was performed by injecting several pulses of reference gas $CO_2$ of known $\delta^{13}C$
value at the beginning and the end of each sample run. Isotopic values are expressed as $\delta^{13}C$
values in per mil relative to the Vienna-PeeDee Belemnite (VPDB). The $\delta^{13}C$ values were
further corrected using a certified isotope standard (Schimmelmann alkane mixture type A6,
Indiana University). The correlation coefficients ($r^2$) of the known $\delta^{13}C$ values of certified
isotope standards with the average values of the measured samples were higher than 0.99. In
the case of silylation of alcohols, we corrected the measured $\delta^{13}C$ values for the isotopic
composition of the methyl adducts (the $\delta^{13}C$ value of the BSTFA = −19.3 ± 0.5 ‰). In order
to monitor the accuracy of the measurements, standards with known $\delta^{13}C$ values were
repeatedly analyzed every 5–6 sample runs. Standard deviations of carbon isotope
measurements were generally better than ±0.4 ‰, as determined by repeated injections of the
standard.

2.6 Genomic DNA extraction and amplification of 16S rRNA genes

Genomic DNA was extracted from 1-2 g of sample using the FastDNA Spin Kit for Soil

(Q-Biogene, Carlsbad, CA, USA). 16S rRNA gene was amplified by polymerase chain
reaction (PCR) using the 8F (3-CTCAGAGTAGTCCGGTTGATCCYGCCGG-5') / 519R
(3'-ACAGAGACGAGGTDTTACCGCGGCKGCTG-5') primers with barcodes for archaeal
community analysis. PCR was carried out with 30 µL of reaction mixture containing
DreamTaq Green PCR Master Mix (2×) (Thermo Fisher Scientific, Waltham, MA, USA), 1
µL of 5 µM primers, and 4 µL of genomic DNA. The PCR procedure included an initial
denaturation step at 94°C for 3 min, 30 cycles of amplification (94°C for 1 min, 55°C for 1
min, and 72°C for 1.5 min), and a final extension step at 72°C for 5 min. Each sample was
amplified in triplicate and pooled. PCR products were purified using the LaboPass
purification kit (Cosmogenetech, Seoul, Korea). Due to PCR failure for samples below 0.6 m
in the MV740 sample, these samples were not included in further analysis.

2.7 Archaeal community and phylogenetic analysis
Sequencing of the 16S rRNA amplicon was carried out by Chun Lab (Seoul, South
Korea) using a 454 GS FLX-Titanium sequencing machine (Roche, Branford, CT, USA).
Preprocessing and denoising were conducted using PyroTrimmer (Oh et al., 2012).
Sequences were processed to remove primer, linker, and barcode sequences. The 3′ ends of
sequences with low quality values were trimmed when the average quality score for a 5-bp
window size was lower than 20. Sequences with ambiguous nucleotides and those shorter
than 250 bp were discarded. Chimeric reads were detected and discarded using the *de novo*
chimera detection algorithm of UCHIME (Edgar et al., 2011). Sequence clustering was
performed using CLUSTOM (Hwang et al., 2013) with a 97 % similarity cutoff. Taxonomic
assignment was conducted for representative sequences of each cluster by EzTaxon-e
database search (Kim et al., 2012). Raw reads were submitted to the National Center for
Biotechnology Information (NCBI) Sequence Read Archive (SRA) database (accession
number PRJNA433786).
For phylogenetic analysis of operational taxonomic units (OTUs) based on 16S rRNA
genes, we selected OTUs belonging to the class *Methanomicrobia* that composed more than
1 % of the relative abundance and aligned them with those of *Methanomicrobia* in jPHYDIT.
A phylogenetic tree was constructed using the maximum-likelihood algorithm (Felsenstein et
al., 1981) with MEGA 6 (Tamura et al., 2013). The robustness of the tree topologies was
assessed by bootstrap analyses based on 1,000 replications of the sequences.

**3 Results**

3.1 Bulk geochemical and microbial lipid analyses

Dissolved sulfate concentrations in sediment cores from MV282, MV420, and MV740 ranged from 0.1 mM to 26.8 mM and sharply decreased within 0.7 m in core depths (Fig. 2, see also Paull et al., 2015). Overall, the TOC contents of core sediments from MV282, MV420 and MV740 ranged from 1.2–1.5 wt.%, 1.0–1.3 wt.%, and 1.1–1.3 wt.%, respectively (Fig. 2, see also Table 1). Similarly, $\delta^{13}C_{TOC}$ values in MV282, MV420 and MV740 cores showed little variation, with average values of –26.3±0.07 ‰, –26.2±0.05 ‰, and –26.3±0.06 ‰, respectively (Fig. 2, see also Table 1).

Isoprenoid dialkyl glycerol diethers (DGDs) archaeol (2,3-di-$O$-phytanyl-$sn$-glycerol) and $sn$-2-hydroxyarcaheol (2-$O$-3-hydroxyphytanyl-3-$O$-phytanyl-sn-glycerol) were identified in the polar fractions of all three cores (Fig. S1); their concentrations were 0.03–0.09 μg/g and 0.01–0.13 μg/g, respectively (Fig. 3, see also Table 1). $Sn$-3-hydroxyarchaeol was identified only in MV282 and MV420 sediments at concentrations of 0.01–0.08 μg/g (Fig. 3, see also Table 1). Among non-isoprenoid DGDs, we identified DGD (If) with *anteiso* pentadecyl moieties attached at both the $sn$-1 and $sn$-2 positions in all three cores. The concentrations of non-isoprenoid DGD (If) ranged from 0.06 to 0.25 μg/g (Fig. 3, see also Table 1). Isoprenoid glycerol dialkyl glycerol tetraethers (GDGTs) containing 0 to 3 cyclopentane moieties (GDGT-0 to GDGT-3) and crenarchaeol which, in addition to 4 cyclopentane moieties, contains a cyclohexane moiety, were detected in all samples investigated (Fig. 4). Overall, the isoprenoidal GDGTs were dominated by GDGT-0 and crenarchaeol, with concentrations of 0.02–0.19 μg/g and 0.02–0.25 μg/g, respectively, whereas GDGT-1 and GDGT-2 showed much lower concentrations (≤0.02 μg/g) in the three cores. In the apolar fractions, we did not detect any isoprenoid hydrocarbons that are typically associated with ANMEs, i.e., the $C_{20}$ compound 2,6,11,15-tetramethylhexadecane (crocetane)

or the $C_{25}$ compound 2,6,10,15,19-pentamethylicosane (PMI).

At the three MVs, the $\delta^{13}C$ values of archaeol and *sn*-2-hydroxyarchaeols ranged from –

79.8 to –38.5 ‰ and from –113.9 to –82.1 ‰, respectively (Fig. 3, Table 1). The $\delta^{13}C$ values
of *sn*-3-hydroxyarchaeol were as low as –93.1 ‰. The $\delta^{13}C$ values of the non-isoprenoid
DGD (If) varied between –46.9 and –31.9 ‰. The $\delta^{13}C$ values of BPs derived from the
isoprenoid GDGTs ranged from –63.4 to –16.7 ‰. The $\delta^{13}C$ values of BP-1 (on average –
51.0 ‰) were slightly more depleted than those of BP-0 (on average –34.2 ‰), BP-2 (on
average –28.3 ‰) and BP-3 (on average –27.5 ‰).

3.2 Depth profile of archaeal classes

Eight different archaeal classes were detected; five of these groups belong to the

Euryarchaeota and three belong to the Crenarchaeota and the Thaumarchaeota (Table S1 and
Fig. S2). The archaeal classes detected were Miscellaneous Crenarchaeotal Group (MCG)_c,
Methanomicrobia, SAGMEGMSBL_c, Thermoplasmata, Marine Benthic Group B
(MBGB)_c, MHVG3_c, Marine Group 1a_c, and Marine Group 1b_c. MCG_c of the phylum
Crenarchaeota was the most dominant archaeal class at the three MVs at a range of depths,
with the exception of the surface of MV420, accounting for 39.7 to 99.2 % of the total
archaeal sequences. In contrast to the archaeal communities below 0.3 m in MV282 and 1.1
m in MV420, which were dominated by MCG_c, shallow archaeal communities at depths of
0.0–0.2 m at MV282, 0.1–0.7 m at MV420, and 0.1–0.6 m at MV740 had different
compositions in the MVs. The class *Methanomicrobia* represented a relatively high
proportion (up to 20.9 %) in these shallow depths at all three MVs.

**4 Discussion**
4.1 Signals of AOM activity in Beaufort Sea mud volcanoes
Active gas bubble emissions into the overlying water column have previously been
observed at all the investigated MVs, i.e., MV282, MV420, and MV740 (Paull et al., 2011
and 2015). A sharp decrease in pore water sulfate concentration and a rapid increase in
sediment temperature near the seafloor indicates the ascension of sulfate-depleted, warm
fluids containing methane from these MVs (Paull et al., 2015). Thus, several lines of
evidence suggest that interstitial methane gas is likely saturated near the seafloor of the
investigated MVs, meaning that both an electron acceptor (sulfate) and a donor (methane) for
AOM are present in the near-surface sediments. Furthermore, an indirect indication of AOM
in near-surface sediments is the presence of thiotrophic organisms, i.e., siboglinid tubeworms
closely related to *Oligobrachia haakonmosbiensis* and the white bacterial mats found at the
summit of MV420 (Paull et al., 2015). Such thiotrophs, which consume the AOM end
product, sulfide, are typically found in habitats characterized by high AOM activity in the
near-surface sediments (Niemann et al., 2006; Rossel et al., 2011; Felden et al., 2014).
AOM at active methane seeps typically proceeds with sulfate as the terminal electron
acceptor (Boetius et al., 2000, Reeburgh, 2007; Knittel et al., 2009; James et al., 2016),
although recent research also found indications for AOM with electron acceptors other than
sulfate, i.e. oxidised Mn and Fe species (Beal et al., 2009) or nitrate/nitrite (Haroon et al.,
2013). The key microbial communities involved in sulfate-dependent AOM are anaerobic
methane oxidisers (ANMEs) in association with sulfate reducing partner bacteria (Knittel et
al., 2009), although ANMEs may also mediate sulfate-dependent AOM without bacterial
partners (Milcuka et al., 2012). AOM with alternative electron acceptors in marine settings is
probably mediated by specialised ANMEs (Beal et al., 2009; Haroon et al., 2013), but it
remains unclear how far potential bacterial partners are involved in these processes. At the
MVs investigated here, we found indications for sulfate-methane transition zones (SMTZ)
because sulfate penetrated only about 0.20 m (MV270), 0.20 m (MV420) and 0.45 m
(MV740) into the sea floor, and we found corresponding elevated abundancies of sulfate-
dependent AOM communities and their lipid biomarkers (Fig.2 and see discussion on AOM
communities in sediments in section 4.2). In contrast to sulfate, the other potential electron
acceptors for AOM mentioned above are typically depleted at shallow depths because redox-
reactions are more thermodynamically feasible than AOM (Reeburgh, 2007). We did not
detect any of the archaeal communities (i.e., *Methanoperedens nitroreducens*, Haroon et al.,
2013) that mediate AOM with electron acceptors other than sulfate, which makes alternative
modes of AOM at the investigated MVs rather unlikely.

4.2 Contribution of AOM to sedimentary biomass
AOM-derived biomass (including lipids) is generally depleted in $^{13}$C compared to the
$\delta^{13}$C-values of source methane as a result of isotopic fractionation during methane
assimilation (Whiticar, 1999). As AOM-related biomarkers, we found substantial amounts of
*sn*-2-hydroxyarchaeol among the isoprenoid DGDs in all three MV sediment cores (Fig. 3).
*Sn*-3-hydroxyarchaeol, an isomer of *sn*-2-hydroxyarchaeol (e.g. Pancost et al., 2000; Elvert et
al., 2005; Niemann et al., 2005; Bradley et al., 2009), was also detected in MV282 but not in
MV420 or MV740, except at 0.7 m in MV420 (Fig. 3). The $\delta^{13}$C values of *sn*-2-
hydroxyarchaeol were more depleted than the $\delta^{13}C_{CH4}$ values (by about –64 ‰, Paull et al.,
2015), with average $\Delta\delta^{13}$C values (lipid-methane) of –35.5 ‰ in MV282, –33.8 ‰ in MV420,
and –29.5 ‰ in MV740. Notably, the $\Delta\delta^{13}$C values of *sn*-2-hydroxyarchaeol were slightly
larger in MV282 than in the other MVs. Similar to *sn*-2-hydroxyarchaeol, the $\delta^{13}$C values of
*sn*-3-hydroxyarchaeol in the MV sediments were generally more depleted than the $\delta^{13}C_{CH4}$
values. Accordingly, the depleted-$\delta^{13}$C values of *sn*-2- and *sn*-3-hydroxyarchaeol indicated
recent AOM occurrence in sediment where sulfate was present. On the other hand, the
depleted $\delta^{13}$C values of *sn*-2-hydroxyarchaeol detected below the SMTZ were likely a fossil

AOM signature (Lee et al., 2013). Non-isoprenoid DGD (If), identified as a potential marker of sulfate-reducing bacteria (SRB) (e.g. Pancost et al., 2001a; Werne et al., 2002), was detected throughout all three MV sediment cores (Fig. 3). However, the $\delta^{13}C$ values of the non-isoprenoid DGD (If) (–46.9 to –32.6 ‰) were enriched in $^{13}C$ relative to the ascending methane in the MVs. Therefore, our $\delta^{13}C$ data from the non-isoprenoid DGD (If) suggest that those compounds originate from a mixed community mediating AOM and other processes.

Furthermore, our measurements of the TOC content and $\delta^{13}C_{TOC}$ values in the three sediment cores revealed narrow ranges of 1.2±0.1 wt.% and –26.4±0.6 ‰, respectively (Fig. 2, see also Table 1), without the negative isotopic excursion that has often been observed in MVs in association with methane-derived biomass from AOM (e.g. Haese et al., 2003; Werne et al., 2004). Therefore, our bulk geochemical data suggest that the contribution of AOM-biomass to sedimentary TOC was rather low at the MVs we investigated, which is in line with our findings that the non-isoprenoid GDGTs substantially originate from bacterial sources unassociated with methanotrophy.

Similarly, we found substantial amounts of archaeal lipids that originated from sources other than AOM. All sediment cores from the three MVs showed a predominance of GDGT-0 and crenarchaeol (Fig. 4), revealing the contribution of marine pelagic Thaumarchaeota (Schouten et al., 2013). The isoprenoid GDGT distributions also did not show a clear dominance of GDGT-2 over GDGT-0. The values of the GDGT-0/crenarchaeol (Liu et al., 2011), the GDGT-2/crenarchaeol (Weijers et al., 2011), and the methane index (Zhang et al., 2011) were also low, with ranges of 0.8–1.7, 0.1–0.2, and 0.2–0.4, respectively. Thus, the GDGT signals found here indicate the negligible contribution of Euryarchaeota to AOM and the GDGT pool (e.g. Pancost et al., 2001b; Zhang et al., 2003; Niemann et al., 2005; Stadnitskaia et al., 2008a, b). The $^{13}C$-enriched isotopic signatures of BPs (Table 1) relative to methane provide further evidence that the isoprenoid GDGTs derived from methanotrophic

archaea were low in the investigated sediments. For example, at sites characterized by high
AOM activity, previous studies found GDGT-1 and -2 at concentrations of up to 20 µg/g,
100-fold higher than in our results (Stadnitskaia et al., 2008b). We can only speculate about
the reasons for the low abundances of AOM-related archaeal communities contributing to the
GDGT pool. One possibility is a rather recent onset in seepage activity at the coring sites,
which would leave too little time for the slow-growing AOM communities, which are
characterized by doubling times on the month scale, to have grown large (Nauhaus et al.,

2007).


4.3 AOM-related microbial communities in Beaufort Sea mud volcanoes
*4.3.1 Chemotaxonomy*
The composition of microbial lipids and their $\delta^{13}C$ values can be used to infer the
chemotaxonomic composition of microbes involved in sulfate-dependent AOM (Niemann
and Elvert, 2008 and references therein). Previously, three groups of anaerobic
methanotrophic archaea (ANME-1, ANME-2 and ANME-3) have been reported in a diversity
of cold seep environments, which are related to methanogens of the orders
*Methanosarcinales* and *Methanomicrobiales* (Knittel and Boetius, 2009). Archaeol is
ubiquitous in archaea, often serving as an indicator of methanogenic archaea in a wide range
of environments including MVs (e.g. De Rosa and Gambacorta, 1988; Koga et al., 1993,
1998; Pancost et al., 2011). In contrast, *sn*-2-hydroxyarchaeol has only been found in certain
orders of methanogens such as *Methanosarcinales, Methanococcales*, *Methanopyrales,*
*Thermoplasmatales, Sulfolobales* and *Methanomicrobiales* (e.g. Kushwaha and Kates, 1978;
Koga et al., 1993, 1998; Koga and Morii, 2005), and s*n*-3-hydroxyarchaeol has been detected
in *Methanosarcinales* (*Methanosaeta concilii*) and *Methanococcales* (*Methanococcus voltae*)
(Ferrante et al., 1988; Sprott et al., 1993).

Microbial communities dominated by ANME-2 at the cold seeps of the northwestern Black Sea contained higher amounts of *sn*-2-hydroxyarchaeol relative to archaeol, whereas the reverse was observed in microbial mats dominated by ANME-1 (Blumenberg et al., 2004). Indeed, the ratio of isotopically depleted *sn*-2-hydroxyarchaeol relative to archaeol can be used to distinguish ANME-1 (0–0.8) from ANME-2 (1.1–5.5), with ANME-3 (2.4) falling within the range of ANME-2 (Niemann et al., 2006; Niemann and Elvert, 2008). In our dataset, the concentration of *sn*-2-hydroxyarchaeol was slightly higher than that of archaeol in MV282, but lower in MV420 and MV740 (Fig. 3, see also Table 1). Accordingly, the *sn*-2-hydroxyarchaeol/archaeol ratio was between 1.3 and 1.8 in MV282, but below 0.7 for most of the samples from MV420 and MV740, except for at depths of 0.7 m (1.4) in MV420 and 0.4–0.6 m (0.9–1.1) in MV740 (Fig. 3, see also Table 1). This observation suggests that ANME-2 (or ANME-3) was involved in AOM in MV282, whereas ANME-1 was probably involved in AOM in MV420 and MV740, except for at the depths mentioned above.

However, the $\delta^{13}$C values of archaeol were on average –62.6 ‰ in MV282, –49.4 ‰ in MV420, and –54.3 ‰ in MV740, except for at 0.7 m in MV420 (–79.8 ‰). Hence, the $\delta^{13}$C values of archaeol in most of the MV sediments appeared to be enriched in $^{13}$C in comparison to that of the ascending methane in the MVs (about –64 ‰, Paull et al., 2015), indicating admixture from processes other than AOM. Hence, it appears that the ratio of *sn*-2-hydroxyarchaeol to archaeol was generally high in all investigated MVs, hinting a negligible involvement of ANME-1 in AOM even in MV420 and MV740. Previous studies showed that GDGTs were mostly absent in ANME-2-dominated settings, but not in ANME-1-dominated settings, which typically contain substantial amounts of GDGT-1 and GDGT-2 (e.g., Blumenberg et al., 2004; Stadnitskaia et al., 2008a, b; Chevalier et al., 2011; Kaneko et al., 2013). The GDGT distributions found here (Fig. 4) indeed show a clear dominance of GDGT-0 and crenarchaeol over GDGT-1 and GDGT-2. Hence, our lipid data indicate that ANME-2

and/or ANME-3 are involved in AOM in the Beaufort Sea MVs rather than ANME-1. We did
not detect crocetane, which is diagnostic for ANME-2 (Elvert et al., 1999), but we also found
no PMIs which are structurally similar to crocetane and produced by ANME-1, -2 and -3
(Niemann and Elvert, 2008), so we could not carry out a further chemotraxonomic distinction
of the dominant ANME groups.

*4.3.2 Nucleic acid based phylogeny*

To further identify key AOM communities, we investigated the archaeal community by

pyrosequencing of 16S rRNA genes. In line with geochemical and biomarker signals for
AOM in the surface sediments of the investigated MVs, we found archaeal sequences of the
*Methanomicrobia*, which contains the order *Methanosarcinales* (i.e., the clade to which the
ANME archaea also belong) at higher abundances in the upper depths of the MV sediment
cores than the lower depths (see Table S2 and Fig. S2). To further clarify the phylogenetic
position within the class *Methanomicrobia* (comprising both methanogens and
methanotrophs), phylogenies of the three most dominant (more than 1 % of all archaeal
sequences) *Methanomicrobia* OTUs (c116, c1698, and c1784) were inferred from 16S rRNA
gene sequences (Supplementary Information Table S2). The OTU c116 represented 2.5–14.1 %
and 0.2–6.7 % of the *archaeal* sequences at core depths of 0.0–0.2 m in MV282 and 0.1–1.1
m in MV420, respectively, whereas this OTU was less than 0.2 % at MV740 (Supplementary
information Table S2). The OTU c1698 accounted for more than 1 % of the archaeal
sequences at the surface of MV282 but was absent at other MVs. The OTU c1784 accounted
for 1.2–6.8 % and 3.7–14.9 % of the archaeal sequences at core depths of 0.0–0.2 m in
MV282 and 0.4–0.6 m in MV740, respectively. In contrast, this OTU was rarely detected at
all depths of MV420, except for at the depth of 0.7 m. The OTUs c116 and c1698 belonged to
ANME-3 archaeal lineage and the OUT c1784 formed a cluster with sequences of ANME-2c,
a distinct lineage of *Methanosarcinales* (Fig. 5). Hence, the occurrence of these sequences,
together with our lipid data, provides evidence that the AOM communities belong to the
ANME-2 and ANME-3 clades; ANME-1 does not seem to play a role at the investigated
Beaufort Sea MVs. In line with our geochemical and lipid analyses, the abundance of
ANME-sequences was also low, underscoring that the contribution of the AOM communities
to the archaeal biomass at the MVs investigated here was rather minor. Instead, we found that
most archaeal sequences belong to the MCG_c clade (up to 99.2 % of all sequences) within
the phylum Crenarchaeota. Although members of this clade were previously shown to
perhaps be involved in methane oxidation in marine and estuary settings (Inagaki et al., 2006;
Jiang et al., 2011; Li et al., 2012), little is known about their physiology and biogeochemical
roles in nature.

4.4 Mechanism controlling microbial communities in Beaufort Sea mud volcanoes
16S rRNA signatures from the Beaufort Sea MVs revealed the presence of AOM related
to archaeal ANME-2 and ANME-3, albeit in relatively low proportions (Fig. 5). The ANME-
2 can be divided into three subgroups, ANME-2a, ANME-2b, and ANME-2c (e.g. Orphan et
al., 2001; Knittel et al., 2005). In the Beaufort Sea MVs, the ANME-2c subgroup was
detected (Fig. 5). A previous study at Hydrate Ridge (Cascadia margin off Oregon, USA)
showed that ANME-2c was dominant at symbiotic clam *Calyptogena* sites, accounting for
>75 % of the total ANME-2, whereas ANME-2a was the most abundant at a side covered by
the sulfide-oxidizing bacterium *Beggiatoa*, accounting for up to 80 % (Knittel et al., 2005).
Fluid flow rates and the methane fluxes from the seafloor were substantially weaker at
*Calyptogena* sites than at *Beggiatoa* sites (e.g. Tryon et al., 1999; Sahling et al., 2002). The
distinct distribution of ANME-2 subgroups might reflect their sulfide tolerance and oxygen
sensitivity (Roalkvam et al., 2011). It appears that ANME-2c has a preferential niche
interacting with chemosynthetic habitats in relatively low methane fluxes in the Beaufort Sea
MVs.

The thermal gradients in our study area (see Paull et al., 2015) were substantially higher

in the MVs (517.7 mK/m in MV282, 557.9 mK/m in MV420, and 104.3 mK/m in MV740)
than in the reference site (28.9 mK/m). In general, high geothermal gradients were observed
where methane emission activities were high, as reported at Dvurechenskii MV (Feseker et
al., 2009) and Haakon Mosby MV (Kaul et al., 2006). Accordingly, among the MV sites, the
methane flux appeared to be the highest at the MV420 site. Indeed, we found a lower
abundance of ANME-2c in MV420 than in MV282 and MV740 (Fig. 5, see also Table S2).
The MV740 site had the lowest thermal gradient of the MV sites, and thus probably the
lowest methane flux, which is consistent with the presence of the gas hydrate flake at 230 cm
in the MV740 sediment core (see Fig. 1D). At this MV site, ANME-2c occurred at a deeper
core depth (0.3–0.7 m) than at the MV282 site (0.0–0.3 m, see also Table S2). This might be
linked to the lower methane flux at the MV740 site than at the MV282 site, resulting in
penetration of sulfate to deeper sediment depths. Notably, at active MV sites, the sulfate
penetration depth can be limited to the upper 2-cm sediment layers (cf. Niemann et al., 2006).

Besides ANME-2c, 16S rRNA gene analyses also revealed the presence of ANME-3 (see

Table S2). Notably, ANME-3 occurred in MV420 whereas thermal gradients were high
(indicating high methane flux) and ANME-2c was almost absent. However, ANME-3 was
absent in MV740 where ANME-2c was present. Similar to ANME-2a, ANME-3 was
previously found at a high fluid flow/methane flux site associated with *Beggiatoa* mats at the
Haakon Mosby Mud Volcano located in Barents Sea at the water depth of 1,250 m (Niemann
et al., 2006, Lösekann et al., 2007). Accordingly, it seems that ANME-3 thrives better in a
setting with higher methane fluxes than ANME-2c.

## 5 Summary and conclusions

Integrated biogeochemical and nucleic acid analyses were performed for three sediment cores retrieved from active MVs in the Beaufort Sea. The sharp decrease in pore water sulfate concentrations and steep thermal gradients and previous observations of gas flare above the edifices indicate that sulfate-depleted, warm fluids and methane ascend from the Beaufort Sea MVs. We found isotopically depleted lipid biomarkers and nucleic acid signatures of microbial communities, most likely ANME-2c and ANME-3, mediating AOM in the surface sediments at these MVs. The prevalence of ANME-3 over ANME-2c at sites characterized by high thermal gradients (and thus probably high methane fluxes) provides a further indication of a methane-flux driven niche segregation of these ANME-clades. However, the overall contribution of AOM-related biomass to the organic carbon pool was rather low, and the presence of dominant amounts of lipid biomarkers with comparably high $\delta^{13}$C-values, as well as the dominance of non-ANME sequences, underscores the importance of processes other than AOM in the sediments of the MVs investigated here. Given that our gravity coring system failed to recover the uppermost surface sediments, preventing us from detecting the most active AOM occurrences in the Beaufort Sea MVs, further studies should investigate the undisturbed uppermost surface sediments to investigate the diversity and distribution of AOM-related archaeal communities in detail, and to clarify their preferred habitats in the Beaufort Sea MV systems, for instance, using ROV push cores.

**Author contribution**

JHK, DHL and YML prepared the manuscript with contributions from AS and HN. DHL, JHK, YML, YKJ, and KHS designed the experiments and were responsible for the analysis. YGK provided thermal gradient data.

**Competing interests**
The authors declare that they have no conflict of interest.

**Acknowledgments**
We would like to thank the captain and crew of R/V ARAON for their safe and skillful
operation of the ship during the cruise. This study was supported by the KOPRI project
(KOPRI-PM18050) funded by the Ministry of Oceans and Fisheries (MOF), and by a
National Research Foundation of Korea (NRF) grants funded by the Ministry of Science and
ICT (MSIT) (NRF-2016M1A5A1901769, KOPRI-PN18081; NRF-2016R1A2B3015388,
KOPRI-PN18100). We also thank Y. Chikaraishi and M. Kaneko for their help in isotope
measurements of biphytanes during a short stay in JAMSTEC, Japan. We are grateful to J.-K.
Gal, S. Kang, and D. Kim for their analytical assistance in the laboratory at Hanyang
University.

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

Index: A tetraether archaeal lipid biomarker indicator for detecting the instability of
marine gas hydrates, Earth Planet. Sci. Lett., 307, 525–534, 2011.

**Table Legends**

Table 1. Results of total organic carbon (TOC) contents, $\delta^{13}C$ of TOC, and concentrations and stable carbon isotopes of selected lipid biomarkers such as isoprenoid DGDs, non-isoprenoid DGDs, and biphytanes derived from isoprenoid GDGTs.

**Figure captions**

Fig. 1. (A) Map showing the study area (red box) with inset regional map of Alaska and northwestern Canada modified from Paull et al. (2015). (B) Map showing the three mud volcano (MV) locations on the upper slope of the Beaufort Sea. (C) Detailed bathymetric maps showing the locations of sediment cores ARA05C-10-GC (MV282), ARA05C-01-GC (MV420), and ARA05C-18-GC (MV740). (D) Lithology of the three sediment cores investigated.

Fig. 2. Depth profiles of sulphate ($SO_4^{2-}$) concentrations, total organic carbon (TOC) content, and $\delta^{13}C_{TOC}$ in sediment cores from MV282, MV420, and MV740. Grey hatched bars indicate gas-gaps in the sediment layers. Note that the sulphate concentration data are from Paull et al. (2015).

Fig. 3. Vertical profiles of selected lipid biomarkers (archaeol, hydroxyarchaeol, and DGD (If)) obtained from sediment cores (A) ARA05C-10-GC (MV282), (B) ARA05C-01-GC (MV420), and (C) ARA05C-18-GC (MV740). Grey hatched bars indicate gas gaps in sediment layers.


Fig. 4. HPLC/APCI-MS base peak chromatograms of polar fractions obtained from sediment
cores (A) ARA05C-10-GC (MV282), (B) ARA05C-01-GC (MV420), and (C) ARA05C-18-
GC (MV740). Note that the Roman numerals (I, II, III, IV and V) refer to GDGT-0, GDGT-1,
GDGT-2, and crenarchaeol, respectively. The Arabic numbers in GDGT-0, GDGT-1,
GDGT-2, and GDGT-3 indicate the number of cyclopentane rings within the biphytane chains.

Fig. 5. Phylogenetic tree based on 16S rRNA showing the relationships of methanomicrobial
sequences recovered in this study with selected reference sequences of the domain
Euryarchaeota. The phylogenetic tree was inferred by the maximum-likelihood method.
Filled circles indicate bootstrap values higher than 70 % based on 1,000 replications. The
scale bar indicates evolutionary distance of 0.05 substitutions per site.

Table 1

| Core depth | TOC | δ¹³C_TOC | Lipid biomarkers | | | | | | | | | | | | | | | | | | hydroxyarchaeol/archaeol |
|---|---|---|---|---|---|---|---|---|---|---|---|---|---|---|---|---|---|---|---|---|---|---|
| | | | Archaeol | | sn-2-hydroxyarchaeol | | sn-3-hydroxyarchaeol | | Non-isoprenoid DGDs | | GDGT-0 | GDGT-1 | GDGT-2 | GDGT-3 | Crenarchaeol | Biphytane 0 | Biphytane 1 | Biphytane 2 | Biphytane 3 | | |
| (mbsf) | (wt. %) | (‰ VPDB) | µg/g dw | ‰ VPDB | µg/g dw | ‰ VPDB | µg/g dw | ‰ VPDB | µg/g dw | ‰ VPDB | µg/g dw | µg/g dw | µg/g dw | µg/g dw | µg/g dw | ‰ VPDB | ‰ VPDB | ‰ VPDB | ‰ VPDB | | |
| **MV282** | | | | | | | | | | | | | | | | | | | | | |
| 0.02 | 1.2 | -26.6 | 0.05 | -65.0 | 0.09 | -107.4 | 0.02 | n.d. | 0.13 | -39.4 | 0.07 | 0.01 | 0.01 | 0.00 | 0.09 | -29.4 | -46.2 | -39.0 | -27.1 | | 1.8 |
| 0.09 | 1.5 | -26.6 | 0.09 | -67.7 | 0.13 | -100.6 | 0.03 | n.d. | 0.15 | -40.1 | 0.07 | 0.01 | 0.01 | 0.01 | 0.08 | -32.3 | -30.9 | -26.8 | | | 1.6 |
| 0.20 | 1.1 | -26.4 | 0.06 | -62.0 | 0.11 | -103.2 | 0.03 | -92.8 | 0.16 | -41.4 | 0.06 | 0.01 | 0.01 | 0.00 | 0.06 | -33.9 | -46.3 | -28.6 | -26.6 | | 1.7 |
| 0.33 | 1.2 | -26.4 | 0.07 | -60.3 | 0.09 | -98.6 | 0.02 | n.d. | 0.16 | -37.9 | 0.05 | 0.01 | 0.01 | 0.00 | 0.05 | - | - | - | - | | 1.3 |
| 0.50 | 1.3 | -26.2 | 0.06 | -64.8 | 0.07 | -99.4 | 0.02 | -84.2 | 0.13 | -45.3 | 0.05 | 0.01 | 0.00 | 0.00 | 0.05 | -36.4 | n.d. | -21.2 | -29.6 | | 1.3 |
| 0.88 | 1.5 | -26.0 | 0.06 | -60.9 | 0.07 | -103.6 | 0.02 | n.d. | 0.12 | -42.9 | 0.05 | 0.01 | 0.01 | 0.00 | 0.05 | - | - | - | - | | 1.3 |
| 1.05 | 1.4 | -26.2 | 0.06 | -60.8 | 0.08 | -91.0 | 0.02 | -87.3 | 0.14 | -36.0 | 0.05 | 0.01 | 0.01 | 0.00 | 0.06 | -32.2 | -36.0 | -29.0 | -16.7 | | 1.5 |
| 1.30 | 1.4 | -26.2 | 0.06 | -63.7 | 0.08 | -97.3 | 0.02 | n.d. | 0.14 | -39.3 | 0.06 | 0.01 | 0.01 | 0.01 | 0.06 | - | - | - | - | | 1.3 |
| 1.60 | 1.5 | -26.0 | 0.06 | -61.8 | 0.08 | -98.1 | 0.02 | -89.1 | 0.14 | -38.8 | 0.05 | 0.01 | 0.00 | 0.00 | 0.05 | -33.5 | -38.6 | -30.7 | -37.6 | | 1.5 |
| 1.90 | 1.2 | -26.5 | 0.03 | -59.2 | 0.03 | -96.0 | 0.01 | n.d. | 0.06 | -43.8 | 0.02 | 0.00 | 0.00 | 0.00 | 0.02 | - | - | - | - | | 1.3 |
| **MV420** | | | | | | | | | | | | | | | | | | | | | |
| 0.08 | 1.0 | -26.4 | 0.03 | -58.6 | 0.02 | -113.8 | n.d. | n.d. | 0.09 | -34.3 | 0.12 | 0.01 | 0.01 | 0.01 | 0.07 | -42.7 | n.d. | n.d. | n.d. | | 0.5 |
| 0.20 | 1.1 | -26.3 | 0.04 | -41.7 | 0.01 | -86.8 | n.d. | n.d. | 0.11 | -36.6 | 0.19 | 0.02 | 0.01 | 0.01 | 0.12 | - | - | - | - | | 0.2 |
| 0.33 | 1.1 | -26.2 | 0.04 | -47.6 | 0.02 | -108.8 | n.d. | n.d. | 0.15 | -31.9 | 0.06 | 0.00 | 0.00 | 0.00 | 0.05 | -34.0 | -61.6 | -22.9 | -24.0 | | 0.6 |
| 0.50 | 1.1 | -26.1 | 0.03 | -38.6 | 0.00 | -94.6 | n.d. | n.d. | 0.11 | -34.3 | 0.03 | 0.00 | 0.00 | 0.00 | 0.02 | - | - | - | - | | 0.1 |
| 0.70 | 1.1 | -26.7 | 0.09 | -79.8 | 0.13 | -113.9 | 0.08 | -93.1 | 0.25 | -46.9 | 0.10 | 0.01 | 0.01 | 0.01 | 0.13 | -30.3 | -51.4 | -28.4 | -24.0 | | 1.4 |
| 0. 88 | 1.2 | -26.2 | 0.06 | -49.0 | 0.03 | -94.7 | n.d. | n.d. | 0.10 | -41.0 | 0.06 | 0.01 | 0.01 | 0.01 | 0.04 | - | - | - | - | | 0.5 |
| 1.05 | 1.2 | -26.0 | 0.06 | -44.2 | 0.03 | -92.4 | 0.02 | n.d. | 0.09 | -40.2 | 0.07 | 0.01 | 0.01 | 0.01 | 0.06 | -32.3 | -55.7 | -30.1 | -30.4 | | 0.5 |
| 1.38 | 1.1 | -26.1 | 0.06 | -45.6 | 0.02 | -95.7 | n.d. | n.d. | 0.11 | -41.5 | 0.07 | 0.01 | 0.01 | 0.01 | 0.06 | - | - | - | - | | 0.3 |
| 1.6 | 1.2 | -26.1 | 0.08 | -45.3 | 0.03 | -97.0 | n.d. | n.d. | 0.11 | -37.2 | 0.08 | 0.01 | 0.01 | 0.01 | 0.07 | -34.8 | -46.8 | -29.4 | -28.2 | | 0.4 |
| 1.81 | 1.2 | -26.0 | 0.07 | -47.7 | 0.03 | -86.4 | n.d. | n.d. | 0.09 | -40.1 | 0.30 | 0.04 | 0.05 | 0.03 | 0.25 | - | - | - | - | | 0.4 |
| 2.17 | 1.3 | -26.1 | 0.06 | -44.8 | 0.02 | -92.1 | n.d. | n.d. | 0.09 | -39.9 | 0.27 | 0.03 | 0.04 | 0.03 | 0.22 | - | - | - | - | | 0.4 |
| **MV740** | | | | | | | | | | | | | | | | | | | | | |
| 0.08 | 1.2 | -26.3 | 0.04 | -38.5 | 0.02 | -86.2 | n.d. | n.d. | 0.11 | -34.3 | 0.07 | 0.01 | 0.01 | 0.01 | 0.05 | -36.2 | -49.5 | -26.8 | -25.3 | | 0.5 |
| 0.20 | 1.1 | -26.3 | 0.04 | -43.6 | 0.02 | -87.8 | n.d. | n.d. | 0.11 | -32.6 | 0.07 | 0.01 | 0.01 | 0.01 | 0.05 | - | - | - | - | | 0.5 |
| 0.35 | 1.3 | -26.4 | 0.05 | -59.6 | 0.05 | -102.4 | n.d. | n.d. | 0.12 | -37.9 | 0.09 | 0.01 | 0.01 | 0.01 | 0.07 | -36.1 | -57.0 | -24.6 | -31.7 | | 0.9 |
| 0. 45 | 1.1 | -26.4 | 0.04 | -69.6 | 0.05 | -103.7 | n.d. | n.d. | 0.12 | -37.5 | 0.09 | 0.01 | 0.01 | 0.01 | 0.06 | -31.5 | -56.0 | -25.2 | -31.4 | | 1.1 |
| 0.55 | 1.2 | -26.5 | 0.05 | -65.1 | 0.05 | -103.3 | n.d. | n.d. | 0.10 | -42.7 | 0.06 | 0.01 | 0.01 | 0.01 | 0.05 | -40.5 | -50.6 | -27.2 | -30.7 | | 1.0 |
| 0. 75 | 1.1 | -26.2 | 0.04 | -58.3 | 0.02 | -93.5 | n.d. | n.d. | 0.11 | -40.4 | 0.08 | 0.01 | 0.01 | 0.01 | 0.05 | - | - | - | - | | 0.7 |
| 1.00 | 1.2 | -26.1 | 0.04 | -59.5 | 0.01 | -93.7 | n.d. | n.d. | 0.09 | -36.7 | 0.09 | 0.01 | 0.01 | 0.01 | 0.07 | -31.0 | -55.5 | -31.4 | -26.8 | | 0.3 |
| 1.13 | 1.1 | -26.2 | 0.03 | -54.4 | 0.01 | -96.7 | n.d. | n.d. | 0.09 | -37.4 | 0.08 | 0.01 | 0.01 | 0.01 | 0.05 | - | - | - | - | | 0.5 |
| 1.55 | 1.3 | -26.4 | 0.03 | -53.9 | 0.01 | -93.0 | n.d. | n.d. | 0.08 | -33.8 | 0.07 | 0.01 | 0.01 | 0.01 | 0.06 | -33.4 | -50.5 | -27.0 | -23.5 | | 0.4 |
| 2.00 | 1.2 | -26.2 | 0.04 | -41.2 | 0.01 | -82.1 | n.d. | n.d. | 0.09 | -35.7 | 0.09 | 0.01 | 0.01 | 0.01 | 0.06 | - | - | - | - | | 0.3 |
| 2.30 | 1.2 | -26.3 | 0.04 | -52.8 | 0.02 | -88.8 | n.d. | n.d. | 0.09 | -38.6 | 0.08 | 0.01 | 0.01 | 0.01 | 0.06 | -34.1 | n.d. | n.d. | n.d. | | 0.4 |
| 2.60 | 1.1 | -26.4 | 0.03 | -55.0 | 0.02 | -90.1 | n.d. | n.d. | 0.08 | -37.3 | 0.06 | 0.01 | 0.01 | 0.01 | 0.05 | - | - | - | - | | 0.5 |



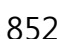

Fig. 1



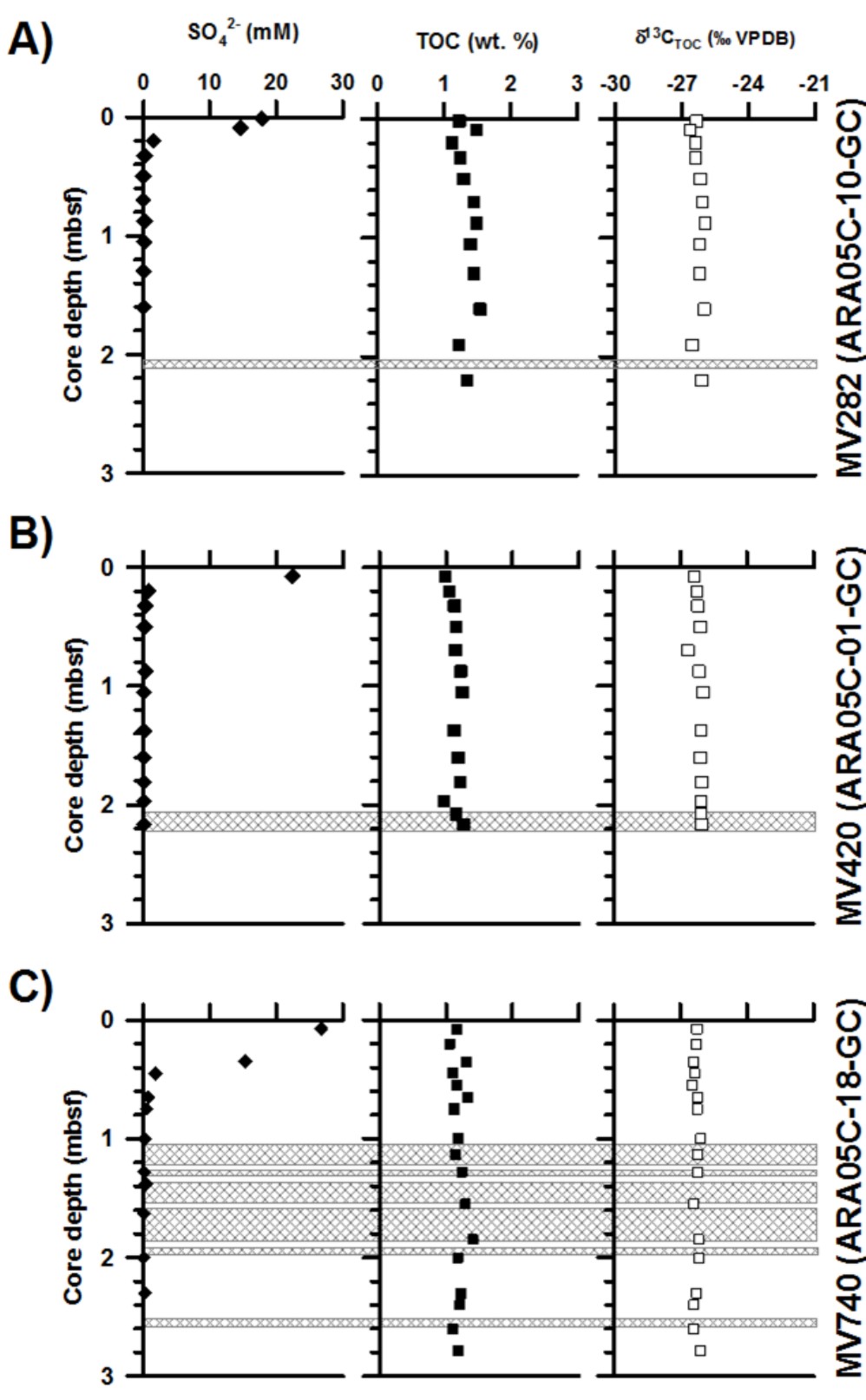

Fig. 2



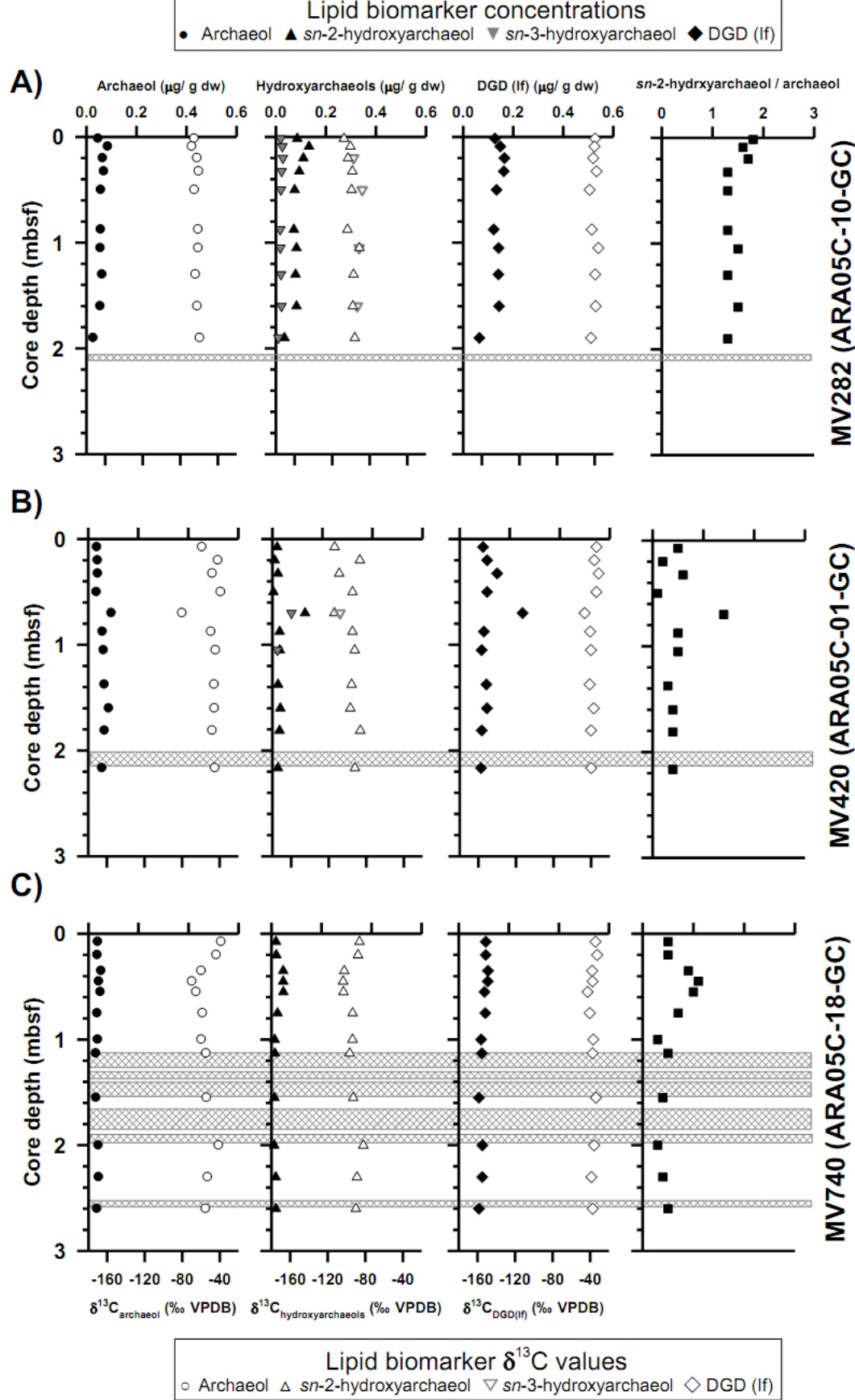

Fig. 3


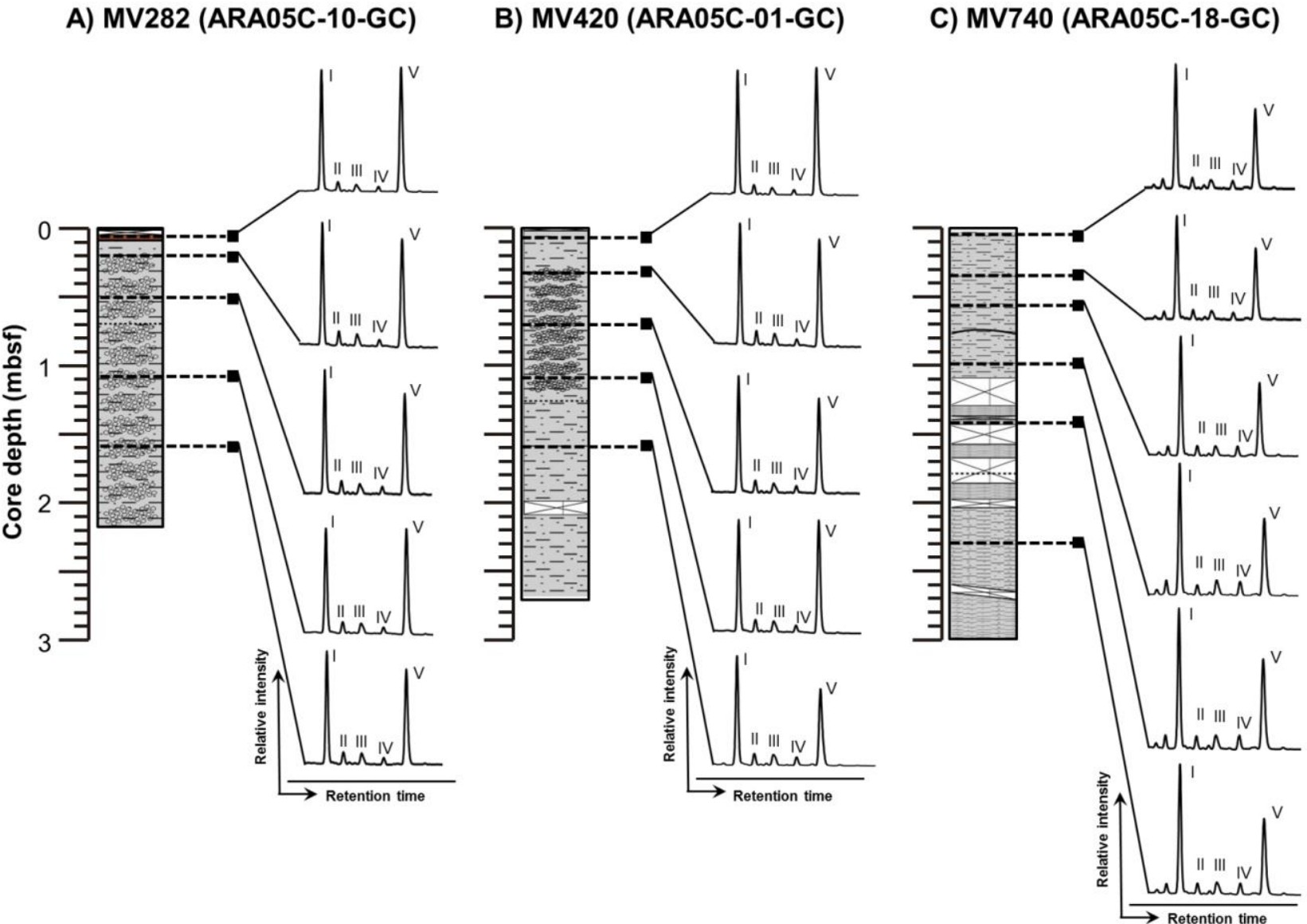

Fig. 4

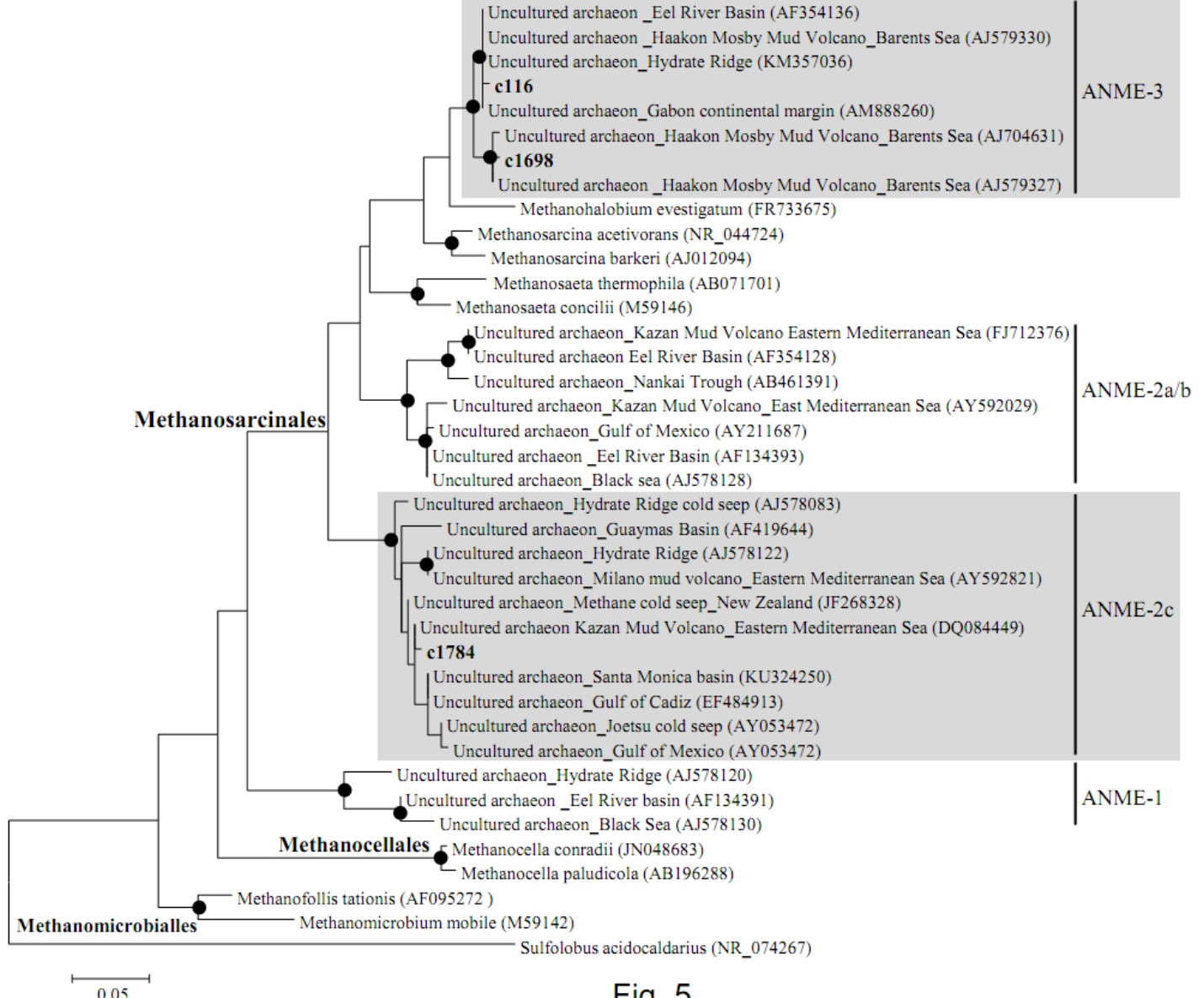


Fig. 5