# Peer review of "Biogeochemical evidence of anaerobic methane oxidation on active submarine mud volcanoes on the continental slope of the Canadian Beaufort Sea"

_Biogeosciences, 2018_

## Referee Comment (RC1) · Anonymous Referee #1 · 14 May 2018

Lee and coworkers investigated the biogeochemistry Pingo-like structures (associated with gas/ mud emissions) in the Beaufort Sea, in particular the imprints of the anaerobic oxidation of methane. In figure 1 these structures are shown, figure 2 shows representative GC-runs , fig. 3 the complete specific results for archaeal lipids. Figure 4 shows results from GDGT analysis, in particular the ring distribution of numbers and a phylogenetic tree of important community members. The concentrations and isotopic compositions of specific compounds as well as (relative) number of 16S sequences are presented in the supplements, although they are by far more interesting

than exemplary GC runs. Hence these data should be displayed in the main text. Although all data seem to be correctly assessed, the discussion is shallow and lacks expertise, which is surprising based on the long and partly well-known author list. Most results from the supplementary table and Supplementary Figure 1 are essential to the study – they show that sulfate methane transition zones are in the upper 20 to 50 cm. Somehow this is not further discussed in the manuscript – although this is core to the biogeochemistry. In summary the authors investigate the imprint of methane in the biochemistry of one of the most remote seas on Earth. Although certainly quiet active - AOM has relatively little impacton total organic carbon contents of the sediment, yet lipid and microbial composition data shows presence and activity of key organisms. The data is there - but the discussion of it needs to be strongly revised. The biogeo-chemistry of AOM as suggested in the title / abstract are -so far - not really covered.

Below I discussed some findings in more detail "Evidence for AOM:" The best evidence for AOM here is the depletion of sulfate, the presence of highly depleted lipids and larger sequence numbers in respective horizons. The rest is not very meaningful "However, organic carbon contents and $\delta$13CTOC values of the three sediment cores investigated spanned a narrow ranges of 1.2±0.1 wt.% and −26.4±0.6‰ respectively (see Table S1 and Fig. S1), without the negative isotopic excursion that has often been associated with methane-282 derived biomass from AOM in MVs (e.g. Haese et al., 2003; Werne et al., 2004)." AC: It is quite normal that the total organic carbon content and the total carbon isotopic composition are only slightly influences – I wonder why the TOC data are shown in the main text but those of specific lipids Abundance and isotopic compositions of GDGTs GDGTs are always very abundant in sediments but they derive with very distinct isotopic compositions around ∼-25 permil from the water column. The Dataset by Lee et al., clearly shows that AOM shifts the GDGT isotopic composition from -25 to around -45 permil. Although this is for sure less than the – 60 permil of methane, it is a clear imprint on the isotopic composition. Those data should be transferred into the main text. Moreover assuming an origin of the ANME derived GDGTs from head to head condensation of archaeol lipids (i.e. c.f. Kellermann et al.,

2016; Org Geochem) one could determine the contribution of archaea to the GDGT pool assuming similar concentrations of both compounds.

Line 320 ff. very important to notice: sulfate is only present in surface sediments, all sediments below (20 to 50 cm) are methanogenic at present time. Lipids particular of archaeal origin are preserved for long. I miss the discussion of data Line 334 ff: Chemotaxonomy: It should not be stated it is strictly ANME-1 or ANME 2, both organisms can exist next to each other. Furthermore, ANME lipids may remain also after in zones without AOM activity. This becomes evident when analyzing the microbial compositions in 4.2.2. Potential AOM zones have the highest sequence numbers of ANME archaea. This should be clearly discussed

4.3. Albeit in the title of this sections, mechanisms controlling the microbial community compositions were not discussed at all. However, the zones of highest ANME sequence numbers are in agreement with AOM zone. Other than this it should be discussed how the other archaeal groups develop with depth.

Conclusions: Nothing new, I am not sure if those conclusions are needed.

Some detailed comments along reading.

Abstract: Please find another start: That sounds very technical (Line 31 to 34): AOM related biomass mainly derives from inorganic carbon (i.e. Kellermann et al., 2012, Wegener et al., 2016 Front. Microbiol), hence this discussion point is rather weak – and of course most biphytanes do not come from methane Lien 31 A value cannot be enriched, but is either high or low Line 49: Why not simpler: The following mapping of the southern Beaufort Sea revealed numerous Lne 54: Based on their formation processes PLFs can be classified into five categories; please also state how they are formed Line 55ff: "The PLFs on the Beaufort Sea shelf appear to be geographically controlled by the presence of submerged permafrost" The appearance of PLFs on the Beaufort Sea is connected / seems to be connected with the presence of permafrost – or something similar? Line 56ff: If PLFs have different origins, please make clear which

one you discuss now. . . are these the true pingos now, and do you stop discussing the other ones from here on? Line 72 indicating microbial production Line 76: The PFAs of the Beaufort Sea are mapped and fluid dynamics have been reasonable well understood, but the biogeochemistry of processes related to the anaerobic oxidation of methane (AOM) were not investigated. Line 82: but the microbial communities involved in the anaerobic oxidation of methane Line 96: "Upon recovery, all three sediment cores were observed to expand and bubble profusely" – rewrite . Upon recovery, in all three sediment cores . . . was observed. Line 97: Start sentence with on board – because you likely sampled on board but did not do the analyses. Line 108: Revise sentence The isotope ratios of TOC were reported in as deviations against the Vienna Pee Dee Belemnite (VPDB) Line 189: how much DNA have you used for PCR, what is the specificity of these primers . . .. Guess it is a primer for the amplification for partial 16S sequences of archaea. Please also reference these primers if you have not developed them Line 205 – 208: please reference tools used for these operations Fig.1 is only later discussed; it should be mentioned earlier, i.e. in Methods, the results introduction to be Fig. 1

There are for sure more details to discuss, but first the MS needs to be revised.

---

## Short Comment (SC1) · 28 May 2018

This paper reports the occurrence of AOM in the active mud volcanos on the continental slope of the Canadian Beaufort Sea, based on the analysis of lipid biomarker and 16S rRNA gene in three sediment cores. The authors state that archaea of the ANME-2c and ANME-3 clades participated in the process of methane consumption, by the relative high ratio of sn-2-hydroxyarchaeol to archaeol and the phylogenetic identities. They then speculate that the difference in distribution of ANME-2c and ANME-3 in these mud volcanos is under the control of methane flux changes. The results of this

study can further help us to understand the biogeochemical cycling of methane in the seafloor mud volcano environments.

Generally, it is a very interesting study, especially the detection of extreme depleted $\delta$13C value of sn-2- and sn-3-hydroxyarchaeol. The approach is scientifically valid and the conclusions are reasonably sound. However the manuscript has some questions and inadequate discussion that require resolution.

1. The content of the abstract section needs some more substance. For example, the mechanism of ANME distribution, as discussed in section 4.3, should be presented in this section.

2.Some expression is inaccurate, such as "bulk elements" at line 101 and "Bulk geochemical analysis" at line 105 . The "Bulk geochemical analysis" only contains the bulk total organic carbon and its carbon stable isotope composition in this manuscript. However, "bulk elements" indicates that the data also contain other chemical parameters such as the major and trace elements of the sediments.

3.In the section 4.1 (Line 292-306), the authors suggest that AOM was mainly coupled with sulfate reduction in this area. However, sulfate became depleted at depth of 0.20 (MV270), 0.20 (MV420) and 0.45m bsf (MV740) indicated by data of Paull et al., 2015. The reason why biomarker of AOM was detected in the sulfate depleted zone in this study need to be clarified.

4.In section 4.3, the authors speculate that the distribution of ANME-2c and ANME-3 is under control of methane flux. Specifically, ANME-2c has a preferential niche in habitats with lower methane fluxes, while ANME-3 is apt to present in setting with relative higher thermal gradients (corresponding to higher methane fluxes). However, ANME-2c and ANME-3 both present in the upper zone (0-0.2m bsf) of sediment core MV282 with similar abundance. Please provide an explanation for this phenomenon.

5. Line 384 to 388, the authors suggest that archaeol are originated from methanogenesis rather than AOM, due to their relative enriched $\delta$13C values (-79.8‰ to -38.5‰ Table S1 ) than the ascending methane in the MVs (about -64 ‰. This explanation seems a little superficial. If archaeol was relate to methanogenesis in this area, the $\delta$13C value of archaeol should be much heavier than those of methane and dissolved inorganic carbon in the methanogenesis zone. However, the difference in $\delta$13C value between archeaol and methane, especially in MV282, seems not large enough to draw this conclusion. Moreover, if this conclusion is correct, the phenomenon that the occurrence of methanogens in the AOM zone should be explained in the manuscript, and references should be added to support this conclusion.

6.Line 390-392, "previous studies...showed that GDGTs were mostly absent in ANME-2-dominanted settings". Please clarify why ANME-2 and GDGTs were both detected in your data.

7.There are some spelling mistakes throughout the manuscript (e.g., Line 284 "Several of the lines of evidence" should be "Several lines of evidence". Line 416 "OUT c1698" should be "OTU c1698").

---

## Short Comment (SC5) · 7 Jun 2018

Short comment #1: Manuscript ID: bg-2018-91 Biogeochemical and microbiological evidence for methane-related archaeal communities at active submarine mud volcanoes on the Canadian Beaufort Sea slope

This paper reports the occurrence of AOM in the active mud volcanos on the continental slope of the Canadian Beaufort Sea, based on the analysis of lipid biomarker and 16S rRNA gene in three sediment cores. The authors state that archaea of the ANME-

2c and ANME-3 clades participated in the process of methane consumption, by the relative high ratio of sn-2-hydroxyarchaeol to archaeol and the phylogenetic identities. They then speculate that the difference in distribution of ANME-2c and ANME-3 in these mud volcanos is under the control of methane flux changes. The results of this study can further help us to understand the biogeochemical cycling of methane in the seafloor mud volcano environments. Generally, it is a very interesting study, especially the detection of extreme depleted $\delta$13C value of sn-2- and sn-3-hydroxyarchaeol. The approach is scientifically valid and the conclusions are reasonably sound.

Reply: We are thankfull for the positive assessment about our manuscript.

However the manuscript has some questions and inadequate discussion that require resolution.

Reply: We will revise our MS as below.

1. The content of the abstract section needs some more substance. For example, the mechanism of ANME distribution, as discussed in section 4.3, should be presented in this section.

Reply: As suggested, we will include the mechanism of ANME distribution in abstract in the revised version.

2. Some expression is inaccurate, such as "bulk elements" at line 101 and "Bulk geochemical analysis" at line 105 . The "Bulk geochemical analysis" only contains the bulk total organic carbon and its carbon stable isotope composition in this manuscript. However, "bulk elements" indicates that the data also contain other chemical parameters such as the major and trace elements of the sediments.

Reply: We will specify it in the revised version.

3. In the section 4.1 (Line 292-306), the authors suggest that AOM was mainly coupled with sulfate reduction in this area. However, sulfate became depleted at depth of 0.20 (MV270), 0.20 (MV420) and 0.45m bsf (MV740) indicated by data of Paull et al., 2015.

The reason why biomarker of AOM was detected in the sulfate depleted zone in this study need to be clarified.

Reply: This is a good point. We will clarify that the archaeal lipids (i.e. AOM related biomarkers) can be preserved for longer period and thus can be present as fossil evidence below the depth of the present SMTZ.

4. In section 4.3, the authors speculate that the distribution of ANME-2c and ANME-3 is under control of methane flux. Specifically, ANME-2c has a preferential niche in habitats with lower methane fluxes, while ANME-3 is apt to present in setting with relative higher thermal gradients (corresponding to higher methane fluxes). However, ANME-2c and ANME-3 both present in the upper zone (0-0.2m bsf) of sediment core MV282 with similar abundance. Please provide an explanation for this phenomenon.

Reply: This is a valid point. Thanks for the nice comment. We are currently not certain why ANME-2c and ANME-3 occurred at the MV282 site. Although we currently think that the distribution of ANME-2c and ANME-3 is probably controlled by the methane flux. But it might be probable that other factors (e.g. salinity) can also influence which is unknown so far. We will address this point in the revised version.

5. Line 384 to 388, the authors suggest that archaeol are originated from methanogenesis rather than AOM, due to their relative enriched $\delta13C$ values (-79.8‰ to -38.5‰ Table S1 ) than the ascending methane in the MVs (about -64 ‰. This explanation seems a little superficial. If archaeol was relate to methanogenesis in this area, the $\delta13C$ value of archaeol should be much heavier than those of methane and dissolved inorganic carbon in the methanogenesis zone. However, the difference in $\delta13C$ value between archeaol and methane, especially in MV282, seems not large enough to draw this conclusion. Moreover, if this conclusion is correct, the phenomenon that the occurrence of methanogens in the AOM zone should be explained in the manuscript, and references should be added to support this conclusion.

Reply: We agree with the reviewer for this point. Although diether lipids (e.g. archaeol)

are less specific and likely produced by all ANMEs and methanogens, as notied by the reviewer, the $\delta$13C value of archaeol is not much heavier than those of methane and dissolved inorganic carbon in the methanogenesis zone. We will revised this part of the discussion in the revised version.

6. Line 390-392, "previous studies...showed that GDGTs were mostly absent in ANME-2-dominanted settings". Please clarify why ANME-2 and GDGTs were both detected in your data.

Reply: Although ANME-2 and GDGTs were both detected in our data, what we ment is that methane-related GDGTs were not dominant in our samples. Thus, we will explain both detected isotopic values and clarify this in the revised version.

7. There are some spelling mistakes throughout the manuscript (e.g., Line 284 "Several of the lines of evidence" should be "Several lines of evidence". Line 416 "OUT c1698" should be "OTU c1698").

Reply: We will correct them in the revised version.

---

## Author Comment (AC1) · 7 Jun 2018

Reviewer #1: Manuscript ID: bg-2018-91 Biogeochemical and microbiological evidence for methane-related archaeal communities at active submarine mud volcanoes on the Canadian Beaufort Sea slope

Lee and coworkers investigated the biogeochemistry Pingo-like structures (associated with gas/ mud emissions) in the Beaufort Sea, in particular the imprints of the anaerobic oxidation of methane. In figure 1 these structures are shown, figure 2 shows

representative GC-runs, fig. 3 the complete specific results for archaeal lipids. Figure 4 shows results from GDGT analysis, in particular the ring distribution of numbers and a phylogenetic tree of important community members. The concentrations and isotopic compositions of specific compounds as well as (relative) number of 16S sequences are presented in the supplements, although they are by far more interesting than exemplary GC runs. Hence these data should be displayed in the main text. Most results from the supplementary table and Supplementary Figure 1 are essential to the study – they show that sulfate methane transition zones are in the upper 20 to 50 cm. Somehow this is not further discussed in the manuscript – although this is core to the biogeochemistry.

Reply: We appreciate for the generally positive opinion of the reviewer about our manuscript. As the reviewer mentioned, we provided all the figures and tables necessary for the manuscript. As suggested by the reviewer, we will move Fig. S1 and Table S1 into the main text in the revised version, but Fig. 2 into the Supplemenatry information as Fig. S1. Although we already mentioned the steep sulfate depletion related to AOM processes in the shallow depths (line 339-342) in the submitted version, we will extend the discussion on the sulfate methane transition zone in the revised version.

Although certainly quiet active - AOM has relatively little impacton total organic carbon contents of the sediment, yet lipid and microbial composition data shows presence and activity of key organisms. The data is there - but the discussion of it needs to be strongly revised. The biogeochemistry of AOM as suggested in the title / abstract are -so far - not really covered.

Reply: The reviewer is right that AOM had a comparably little impact on the TOC content but this is not unusual for AOM settings. However, we do not agree with the reviewer for the point mentioned above. In our opinion, we discussed the AOM presence based on the bulk/biomarker and microbial composition in the submitted version. However, we will strengthen this part of discussion extending the biogeochemistry of AOM in the revised version. For example, we will incorporate a new aspect connecting relatively depleted $\delta$13C values of archaeol and biphytanes with the activity of chemoorganoautoprophs.

Below I discussed some findings in more detail "Evidence for AOM:" The best evidence for AOM here is the depletion of sulfate, the presence of highly depleted lipids and larger sequence numbers in respective horizons. The rest is not very meaningful"However, organic carbon contents and $\delta$13CTOC values of the three sediment cores investigated spanned a narrow ranges of 1.2$\pm$0.1 wt.% and –26.4$\pm$0.6‰ respectively (see Table S1 and Fig. S1), without the negative isotopic excursion that has often been associated with methane-derived biomass from AOM in MVs (e.g. Haese et al., 2003; Werne et al., 2004)." AC: It is quite normal that the total organic carbon content and the total carbon isotopic composition are only slightly influences. I wonder why the TOC data are shown in the main text but those of specific lipids Abundance and isotopic compositions of GDGTs are always very abundant in sediments but they derive with very distinct isotopic compositions around âĹij-25 permil from the water column. The Dataset by Lee et al., clearly shows that AOM shifts the GDGT isotopic composition from -25 to around -45 permil. Although this is for sure less than the – 60 permil of methane, it is a clear imprint on the isotopic composition. Those data should be transferred into the main text. Moreover assuming an origin of the ANME derived GDGTs from head to head condensation of archaeol lipids (i.e. c.f. Kellermann et al., 2016; Org Geochem) one could determine the contribution of archaea to the GDGT pool assuming similar concentrations of both compounds.

Reply: As mentioned above, we will show Table S1 in the main text which includes the concentration and isotopic values of specific lipid biomarkers. As the reviewer notified above, our bulk data showed relatively little impacton of AOM on total organic carbon contents of the sediment and the total carbon isotopic composition. But lipid and microbial data showed the presence of AOM in our study sites. Although diether lipids (e.g. archaeol) are less specific and likely produced by all ANMEs, we will try

to more in detail discuss the contribution of archaea to the GDGT pool in the revised version as suggested by the Reviwer.

Line 320 ff. very important to notice: sulfate is only present in surface sediments, all sediments below (20 to 50 cm) are methanogenic at present time. Lipids particular of archaeal origin are preserved for long. I miss the discussion of data Line 334 ff:

Reply: It is a good point that the archaeal lipid can be preserved for longer period and thus can be present after the present SMTZ, as also mentioned below by the reviewer. This point will be discussed in the revised version.

Chemotaxonomy: It should not be stated it is strictly ANME-1 or ANME 2, both organisms can exist next to each other. Furthermore, ANME lipids may remain also after in zones without AOM activity. This becomes evident when analyzing the microbial compositions in 4.2.2. Potential AOM zones have the highest sequence numbers of ANME archaea. This should be clearly discussed

Reply: We will revise it as recommended by the reviewer.

4.3. Albeit in the title of this sections, mechanisms controlling the microbial community compositions were not discussed at all. However, the zones of highest ANME sequence numbers are in agreement with AOM zone. Other than this it should be discussed how the other archaeal groups develop with depth.

Reply: We agree with the reviewer that the 16S rRNA sequence covers only ANME and thus other archaeal groups are not fully considered. In the revised version, we will incorporate the distribution of other archaeal groups, inferred from predominant GDGTs profiles (e.g. GDGT-0 and crenarchaeol) as well as 16S rRNA sequences. This may hint how the other archaeal groups develop with depth.

Conclusions: Nothing new, I am not sure if those conclusions are needed.

Reply: Our study provides the first biogeochemical and microbial data of active mud volcanoes in the Canadian Beaufort Sea, which enhance our understanding to better

understand methane cycles in this system. Hence, we do think that what stated in the conclusions is valid. In future, a follow-up study will be conducted to examine in more detail distinct methane oxidation processes (e.g. AOM and MOx) near surface using ROV push cores. Nevertheless, we agree that the conclusions also contain a wrapup/summary style of data and will thus rename the section to summary and conclusions.

Some detailed comments along reading. Abstract: Please find another start: That sounds very technical (Line 31 to 34): AOM related biomass mainly derives from inorganic carbon (i.e. Kellermann et al., 2012, Wegener et al., 2016 Front. Microbiol), hence this discussion point is rather weak and of course most biphytanes do not come from methane

Reply: Several studies including the ones stated by the reviewer could show that AOM communities assimilate DIC, however, in AOM systems, this derives from methane. Thus, AOM-derived lipids (and other biomass) is typically 13C-depleted. As mentioned above, we will emphasize the additionally carbon assimilation of AOM-related archaea along the sharply depleted sulfate profiles.

Line 31 A value cannot be enriched, but is either high or low

Reply: We will correct it (as high values).

Line 49: Why not simpler: The following mapping of the southern Beaufort Sea revealed numerous

Reply: As recommended by review, we will correct it (line 51-53) like "The following mapping of the southern Beaufort Sea revealed numerous PLFs which had the various shaped-features may indicate localized unstable seabed conditions."

Line 54: Based on their formation processes PLFs can be classified into five categories; please also state how they are formed

Reply: In our opinion, it is somewhat out of focus of our manuscript. However, we will

briefly mention the formation processes of PLFs in the revised version.

Line 55ff: "The PLFs on the Beaufort Sea shelf appear to be geographically controlled by the presence of submerged permafrost" The appearance of PLFs on the Beaufort Sea is connected / seems to be connected with the presence of permafrost –or something similar?

Reply: As recommended by the reviewer, we will correct it as follows: "The appearance of PLFs in the Beaufort Sea seems to be connected with the presence of submerged permafrost.

Line 56ff: If PLFs have different origins, please make clear which one you discuss now. Are these the true pingos now, and do you stop discussing the other ones from here on?

Reply: The PLFs investigated are the true pingos which are designated as active mud volcanos in the previous studies (e.g. Paull et al., 2007).

Line 72 indicating microbial production

Reply: We will correct this sentence as recommended.

Line 76: The PLFs of the Beaufort Sea are mapped and fluid dynamics have been reasonable well understood, but the biogeochemistry of processes related to the anaerobic oxidation of methane (AOM) were not investigated.

Reply: We will change it as recommended.

Line 82: but the microbial communities involved in the anaerobic oxidation of methane

Reply: We will revise it as follows: "Bulk chemical compounds, specific archaeal lipids and microbial communities were investigated in order to characterize the anaerobic oxidation of methane (AOM) communities in the MV sediments."

Line 96: "Upon recovery, all three sediment cores were observed to expand and bubble

profusely" – rewrite . Upon recovery, in all three sediment cores . . . was observed.

Reply: We will correct this sentence as follows: "Upon recovery, in all three sediment cores, the formation of bubbles was observed in the sediment matrix."

Line 97: Start sentence with on board – because you likely sampled on board but did not do the analyses.

Reply: We will correct it as suggested.

Line 108: Revise sentence The isotope ratios of TOC were reported in as deviations against the Vienna Pee Dee Belemnite (VPDB)

Reply: We will correct it as suggested.

Line 189: how much DNA have you used for PCR, what is the specificity of these primers . . .. Guess it is a primer for the amplification for partial 16S sequences of archaea. Please also reference these primers if you have not developed them

Reply: We will add the quantity of gDNAs used for PCR. Regarding the archaeal primers used in this study, we have developed archaeal primers and that's why only the primer sequences were mentioned without reference. In the revised version of the MS, we will state this more clearly.

Line 205 – 208: please reference tools used for these operations Fig.1 is only later discussed; it should be mentioned earlier, i.e. in Methods, the results introduction to be Fig. 1

Reply: We will mention the Fig. 1 earlier (e.g. in the methods).

---

## Editor Comment (EC1) · Z Jia (Editor) · 8 Jul 2018

This is an interesting study. However, I side with both reviewers, and particularly reviewer #1 that the manuscript is not well organized, although it has potential of being published. The manuscript cannot be published in the current form. However, the re-submission is encouraged.

The writing style is poor and the manuscript needs to be completed re-structured including tables and figures.
If the authors want to make a re-submission, he/she may first revise the tables and figures and send it to me for comments before starting the writing of main text.

Specific comments

1. Title

(1) The title might be improved because it does not specify the role of archaeal communities. Is it methan-metabolizing or something else? There are lots of functions of methane-related archaeal communities. Please get it more focused.
(2) It might be rephrased as "biogeochemical evidence for anaerobic methane oxidation at active submarine mud volcanoes on the Canadian Beaufort Sea slope". The term "Biogeochemical" already contains the meaning of microbiology. In addition, this study is mainly focused on ANME and it is not necessarily extended to "methane-related". Biogeochemical evidence is a mere evidence which does not preclude the importance of other organisms.

2. The abstract needs to be re-organized.

(1) For example, the authors summarized the key findings as the following. "In this study, we provide first evidence of lipid biomarker patterns and phylogenetic identities of key microbes mediating anaerobic oxidation of methane (AOM) communities in active mud volcanos (MVs) on the continental slope of the Canadian Beaufort Sea. Our lipid and 16S rRNA results indicate that archaea of the ANME-2c and ANME-3 clades are involved in AOM in the MVs investigated."
In the abstract, the authors need to present the first evidence of lipid biomarker for anaerobic methane oxidation, and explain why these biomarker can be used. Then the phylogenetic identities of key microbes can be followed. The implication could be then presented.
(2) L31. The enriched 13C. The value cannot be enriched.
(3) L31 to 34. It is a bit unusual to show the data in this way. This is not an important point. The authors claimed that contribution of AOM-related biomass to sedimentary TOC was in general negligible. This might be important, but it is not the key point as specified in the significance section. The number 1 priority is to show the evidence for the presence of AOM, and then microbial identity, and then finish the ms by concluding the importance of AOM biomass contribution.
(4) L35-36. It is a bit unusual to show the evidence in this manner by claiming that "However, the δ 13 C values of sn-2- and sn-3-hydroxyarchaeol were more negative

than CH4, indicating the presence of AOM communities, albeit in a small amount". Firstly, why n-2- and sn-3-hydroxyarchaeol can be used as a biomarker. Secondly, how negative it is, how small the amount it is. This is the key information of this study.

(5) L36-38. This sentence is just the conclusion. The reader need to know the data and evidence, i.e., what is the specific evidence, how the ratio is changing, and why the author feel that the ratio of sn-2- hydroxyarchaeol to archaeol and the 16S rRNA results indeed indicated that archaea of the ANME-2c and ANME-3 clades were involved in AOM.

(6) L38-40. This study already revealed the phylogenetic diversity of AOM, and why future studies are still needed? In addition, why uppermost surface sediments is mentioned, and what is the point?

3. The Introduction

(7) L47. Please delete e.g.

(8) L48. Pls delete the following. by hydrographers aboard the C.C.G.S. John A. MacDonald, a Canadian Coast Guard icebreaker.

(9) L61. Delete e.g.

(10) L69. Delete e.g.,

(11) L74. It needs to be specified why δ13 C CH4 values of −64 ‰, indicating a microbial source

(12) L75. What is the connection between the L74-75 sentence and L76 sentence? It seems rather descriptive

(13) L77. How well it is investigated, what is the key findings about the methane-rich fluid dynamics ?

(14) L78. Please state why this investigation is important, instead of saying that it have not yet been investigated

(15) In summary. Significant revision needs to be made including (a) why AOM could be important in the samples tested in this study; (b) what is the key biomarkers of AOM, and its applicability in this study. For example, the use of GDGT and other archaeol as biomarker for AOM and other archaeal. Maybe the difference in GDGT between ammonia-oxidizing archaea and AOM should be specified; (3) What is the phylogeny of AOM, and what is the expected output of AOM in this study; (4) What is significance if the AOM metabolism is deciphered and so on

4. Materials and Methods

(16) What does the term "methanomicrobial operational taxonomic units" mean?

(17) The authors need to specify how AOM sequences were selected, aligned and analyzed. Of particular concern is the robustness of the phylogenetic identity of AOM

5. Results

(18) L227. Delete the start sentence.

(19) L227-231. Please start the result section with the most important data. It is unusual that

the starting evidence can be placed within the supplementary materials and methods. This TOC is placed in the abstract as the starting point, but why the key data is in supplementary table S1? In case that the author feel TOC is not the most important data, then the most important one should be described first, instead of TOC which can still be used in supplementary table S1.

(20) L299. What does the systematically mean, it can be deleted.

(21)

(22) L258-259. There is no diversity information in Table S2 and Fig. S2. The diversity index is missing. In addition, the majority of sequences in Table S2 and Fig. S2 are from archaea, why it is low?

(23) L260-271. The result section needs to be improved significantly. The current version is somehow pointless. This study is aimed to anaerobic methane oxidation. But only a very small fraction of archaeal communities can be classified as ANME. Whatever, the authors first of all need to emphasize the ANME sequences, then sulfate-reducing sequence, then other sequences.    Pls stay focused on your main theme of this study. Among 25 profile sample detected, apparently Marine Crenarchaeota Group (MCG) predominate archaeal communities in this study except for MV420-0.08. In addition, the authors need to specify the relationship of archaeal lipids to archaea, i.e., what is the specific archaeal lipid for each dominant group of archaeal communities (at phylogenetical level)

6. Discussion

(24) L274. The evidence of AOM in Beaufort Sea mud volcanoes. This title is more appropriate as the result section

(25) L275-277. The authors need to provide the concentration of methane, which is the core data of this study. The data cannot be found in the Fig.1. In addition, simply judged from the title of Paull et al., 2015, it appears that this paper is not closely related to methane

(26) L279. The authors need to specify how the gas is charged in Fig. 1D. please specify

(27) L279-281. What is the relationship of AOM to the fact conveyed by these sentences?

(28) L283-285. What is the point of the interstitial gas?

(29) L285-290. This is not the key point in this study. This conclusion is of minor concern for this study. The contribution of AOM to TOC apparently is out of the scope of this study. In addition, It is also very hard to conclude that the contribution of AOM-related biomass to sedimentary TOC is rather low at the MVs investigated.

(30) L291. The authors first of all need to show methane data

(31) L304. How do the authors know these are sulfate-dependent AOM. If the abundance of sulfate-dependent AOM is elevated, it should be placed in the main text.

(32) L291-305. These sentences are mostly pointless. The evidence of AOM IN THIS STUDY should be first emphasized, and then discussed in the context of other studies.

(33) L308-309. This can be placed in the introduction section.

(34) L311-312. To what extent, the author are certain that these DGD can represent sulfate-reducing bacteria?

(35) L313-314. If it is not supportive of SRB, it may suggest that other electronic acceptor such as Mn/iron/nitrate might be involved? Whatever, it cannot be stopped here and further discussion should be made.

(36) L319. Do the author mean the contribution to GDGD, and so what?

(37) L327-328. There is no solid evidence in support of this statement.

(38) L327-338. Much of these discussion appears more appropriate as the results

(39) L339-341. What is the logic between the sulfate profiles and siboglinid tubeworms???

(40) L344. What does the constrained mean?

(41) L344-350. Maybe the author want to emphasize how sulfate is generated, and then used in support of methane oxidation. The paragraph needs to be re-organized.

(42) L354-356. This should be placed in the introduction

(43) L354-375. Please clearly specify the lipids that are representative of different archaea.

(44) L380. What is ANME2-specific lipid?

(45) L407. To clarify

(46) L407-412. It can be described in the materials and method section

(47) L412-424. These sentences are rather descriptive, and it might be more appropriate in the section of Results.

(48) L430-437. What is your conclusion about AOM in this study, when compared to other studies?

(49) L437. Does this mean that in this study methane concentration is low?

(50) L444 and L450. Where is the measured data of methane flux

(51) L454-464. Part of this discussion should be made in the result section.

(52) L476-477. The authors need to specify that these sn-2- and sn-3-hydroxyarchaeol are representative of AOM at the very beginning.

(53) L480. There is no evidence of methane concentration. How could the author claim that methane was oxidized?

As for the Tables and Figures, they need to be significantly re-structured.

(1) Fig. 1 should be re-organized. Fig. 1B could be placed at the bottom left. Fig. C could be place at top right. Fig. 1D appears to be the most important one which could have more space like the current Fig. 1B, and place in the middle right.

(2) Fig. 2. In a scientific paper it is unusual to show an example figure. This figure tells the readers very little information, and it should be placed in the supplementary, or it should be placed side by side with the data measurements.

(3) The title of this ms is about biogeochemical. Therefore, in the main text, the BIO and the Geochemical data should be included. But all figures are about the BIO evidence.

(4) Table S1 should be placed in the main text as the figure 2. Methane concentration is of particular concern, and should be placed together with sulfate gradient.

(5) The most important data that are related to AOM in Table S1 should be made as a figure and placed in the main text.

(6) What are the key information of Fig. 3 and Fig .4. these data appear to be from Table S1. Please stay focused on the AOM as much as possible.

(54)

---

## Author Comment (AC2) · 17 Jul 2018

Associated editor Manuscript ID: bg-2018-91 Biogeochemical and microbiological evidence for methane-related archaeal communities at active submarine mud volcanoes on the Canadian Beaufort Sea slope

This is an interesting study. However, I side with both reviewers, and particularly reviewer #1 that the manuscript is not well organized, although it has potential of being published. The manuscript cannot be published in the current form. However, the

re-submission is encouraged. The writing style is poor and the manuscript needs to be completed re-structured including tables and figures. If the authors want to make a re-submission, he/she may first revise the tables and figures and send it to me for comments before starting the writing of main text.

Reply: We thank the editor for providing us an opportunity to revise this manuscript. We will revise the manuscript according to the comments made by both the reviewers and the associate editor as indicated in the rebuttals.

Specific comments

1. Title The title might be improved because it does not specify the role of archaeal communities. Is it methan-metabolizing or something else? There are lots of functions of methane-related archaeal communities. Please get it more focused. It might be rephrased as "biogeochemical evidence for anaerobic methane oxidation at active submarine mud volcanoes on the Canadian Beaufort Sea slope". The term "Biogeochemical" already contains the meaning of microbiology. In addition, this study is mainly focused on ANME and it is not necessarily extended to "methane-related". Biogeochemical evidence is a mere evidence which does not preclude the importance of other organisms.

Reply: We will revise the title as suggested by the editor: "Biogeochemical evidence for anaerobic methane oxidation at active submarine mud volcanoes on the Canadian Beaufort Sea slope".

2. The abstract needs to be re-organized. For example, the authors summarized the key findings as the following. "In this study, we provide first evidence of lipid biomarker patterns and phylogenetic identities of key microbes mediating anaerobic oxidation of methane (AOM) communities in active mud volcanos (MVs) on the continental slope of the Canadian Beaufort Sea. Our lipid and 16S rRNA results indicate that archaea of the ANME-2c and ANME-3 clades are involved in AOM in the MVs investigated." In the abstract, the authors need to present the first evidence of lipid biomarker for

anaerobic methane oxidation, and explain why these biomarker can be used. Then the phylogenetic identities of key microbes can be followed. The implication could be then presented.

Reply: We will revise the abstract as suggested by the editor highlighting that the presence and abundance of dignostic lipids allowed this conclusion

L31. The enriched 13C. The value cannot be enriched. Reply: We will correct it as "high values". But we should mention that the isotopic composition can be enriched in 13C.

L31 to 34. It is a bit unusual to show the data in this way. This is not an important point. The authors claimed that contribution of AOM-related biomass to sedimentary TOC was in general negligible. This might be important, but it is not the key point as specified in the significance section. The number 1 priority is to show the evidence for the presence of AOM, and then microbial identity, and then finish the ms by concluding the importance of AOM biomass contribution.

Reply: This is a good point and we will re-structure the abstract following the suggestion made by the editor.

L35-36. It is a bit unusual to show the evidence in this manner by claiming that "However, the $\delta$ 13 C values of sn-2- and sn-3-hydroxyarchaeol were more negative than CH4, indicating the presence of AOM communities, albeit in a small amount". Firstly, why n-2- and sn-3-hydroxyarchaeol can be used as a biomarker. Secondly, how negative it is, how small the amount it is. This is the key information of this study.

Reply: We will revise it as suggested by the editor.

L36-38. This sentence is just the conclusion. The reader need to know the data and evidence, i.e., what is the specific evidence, how the ratio is changing, and why the author feel that the ratio of sn-2- hydroxyarchaeol to archaeol and the 16S rRNA results indeed indicated that archaea of the ANME-2c and ANME-3 clades were involved in

AOM.

Reply: We will add information on the ratio of sn-2- hydroxyarchaeol to archaeol and the 16S rRNA results to draw the conclusion.

L38-40. This study already revealed the phylogenetic diversity of AOM, and why future studies are still needed? In addition, why uppermost surface sediments is mentioned, and what is the point?

Reply: Our study provides the first biogeochemical and microbial data from active mud volcanoes in the Canadian Beaufort Sea. However we could not investigate the oxic-related methanotrophs in this study. That is why further studies are necessary. We will clarify this aspect in the revised version.

The Introduction

L47. Please delete e.g.

Reply: We will delete it.

L48. Pls delete the following. by hydrographers aboard the C.C.G.S. John A. MacDonald, a Canadian Coast Guard icebreaker.

Reply: We will delete it.

L61. Delete e.g.

Reply: We will delete it.

L69. Delete e.g.,

Reply: We will delete it.

L74. It needs to be specified why $\delta$13C CH4 values of –64 ‰ indicating a microbial source

Reply: We will add the aspect of biogenic methane production and migration associated with the microbial signature of d13C value of CH4 (Whiticar, 1999) in the revised version.

L75. What is the connection between the L74-75 sentence and L76 sentence? It seems rather descriptive

Reply: We will clarify the connection between the chemosynthetic communities and the ascending methane source in the Beaufort Sea based on the previous studies.

L77. How well it is investigated, what is the key findings about the methane-rich fluid dynamicsïij§

Reply: Paull et al. (2015) reported that the ascending methane sources might be related to the dissociation of permafrost and/or gas hydrates in subsurface, confirmed by a detailed bathymetric mapping with AUV and seismic survey. Furthermore, previous studies with ROV showed that Beaufort MVs were active edifices characterized by ongoing eruptions. Particularly, the ascending fluid sources can provide essential energy for inhabited microbial organisms's nutritional metabolism. We will clarify this point to strengthen our objectives in the revised vision.

L78. Please state why this investigation is important, instead of saying that it have not yet been investigated

Reply: We will state importancy of AOM reaction regarded as the major barrier against methane efflux from marine sediments into the ocean. We will clearly address it in the revised version.

In summary. Significant revision needs to be made including (a) why AOM could be important in the samples tested in this study; (b) what is the key biomarkers of AOM, and its applicability in this study. For example, the use of GDGT and other archaeol as biomarker for AOM and other archaeal. Maybe the difference in GDGT between ammonia-oxidizing archaea and AOM should be specified; (3) What is the phylogeny of AOM, and what is the expected output of AOM in this study; (4) What is significance

if the AOM metabolism is deciphered and so on

Reply: We agree with the editor for this structure. Thus, we will clarify these parts into the discussion in the revised version.

Materials and Methods

What does the term “methanomicrobial operational taxonomic units” mean?

Reply: “methanomicrobial operational taxonomic units” means the operational taxonomic units of the class Methanomicrobia. We will add more details in the revised manuscript to avoid any confusion.

The authors need to specify how AOM sequences were selected, aligned and analyzed. Of particular concern is the robustness of the phylogenetic identity of AOM

Reply: Sequences of Methanomicrobia which include the archaeal group involved in AOM were selected for the phylogenetic analysis based on their proportion and robusteness of tree topology assesed. Although we already described it, we will clarify it in the revised version.

Results

L227. Delete the start sentence.

Reply: We will delete it as recommended by the editor.

L227-231. Please start the result section with the most important data. It is unusual that the starting evidence can be placed within the supplementary materials and methods. This TOC is placed in the abstract as the starting point, but why the key data is in supplementary table S1? In case that the author feel TOC is not the most important data, then the most important one should be described first, instead of TOC which can still be used in supplementary table S1.

Reply: As also suggested by the reviewer1, we will move the Fig. S1 (in submitted

version) will be shown as a main figure in the revised version. We will also move Table S1 into the main text in the revised version. Instead Fig. 2 (in submitted version) will be moved into the Supplemenatry information as Fig. S1.

L299. What does the systematically mean, it can be deleted.

Reply: We will delete it.

L258-259. There is no diversity information in Table S2 and Fig. S2. The diversity index is missing. In addition, the majority of sequences in Table S2 and Fig. S2 are from archaea, why it is low?

Reply: We agree with the editor's comments since there is no diversity information provided in the submitted verison. The comparison of archaeal diversity is not the main focus of this study and thus we will delete this sentence (L259-260) in the revised version. Instead, the statistics of the sequnces including the diversity indices will be added as a new supplementary table.

L260-271. The result section needs to be improved significantly. The current version is somehow pointless. This study is aimed to anaerobic methane oxidation. But only a very small fraction of archaeal communities can be classified as ANME. Whatever, the authors first of all need to emphasize the ANME sequences, then sulfate-reducing sequence, then other sequences. Pls stay focused on your main theme of this study. Among 25 profile sample detected, apparently Marine Crenarchaeota Group (MCG) predominate archaeal communities in this study except for MV420-0.08. In addition, the authors need to specify the relationship of archaeal lipids to archaea, i.e., what is the specific archaeal lipid for each dominant group of archaeal communities (at phylogenetical level)

Reply: We agree with the editor that the 16S rRNA sequence covers only ANME and thus other archaeal groups are not fully considered. In the revised version, we will incorporate the distribution of other archaeal groups. This may hint how the other

archaeal groups develop with depth.

Discussion

L274. The evidence of AOM in Beaufort Sea mud volcanoes. This title is more appropriate as the result section

Reply: We will revise it as recommended.

L275-277. The authors need to provide the concentration of methane, which is the core data of this study. The data cannot be found in the Fig. 1. In addition, simply judged from the title of Paull et al., 2015, it appears that this paper is not closely related to methane

Reply: Unfortunately, the methane concentration data was not available for this study due to active the gas expansion during core recovery. This manuscript focuses on the AOM-related methanotrophs occurring in three mud volcanoes in the Canadian Beaufort Sea. Particularly, these sites are closely interacting with ascending methane for fueling specific microbial communities (i. e. methanotrophs). Thus, the evidence on gas fluid sources in the previous study (Paull et al., 2015) is important for understanding AOM process by methanotroph.

L279. The authors need to specify how the gas is charged in Fig. 1D. please specify

Reply: In our opinion, this point is somewhat out of focus of our manuscript. However, we will briefly explain it, as suggested.

L279-281. What is the relationship of AOM to the fact conveyed by these sentences? L283-285. What is the point of the interstitial gas?

Reply: The problem of free gas is always that it is not accessible by microbes. And, in most cases, mousse/foamy appearance of sediments is a post sampling artefacet because of gas expansion during recovery. The part of gas description within core sediments will move to the sample collection part in order to clarify AOM evidence in

the discussion.

L285-290. This is not the key point in this study. This conclusion is of minor concern for this study. The contribution of AOM to TOC apparently is out of the scope of this study. In addition, It is also very hard to conclude that the contribution of AOM-related biomass to sedimentary TOC is rather low at the MVs investigated.

Reply: AOM had comparably little impact on the TOC content which is not unusual for AOM settings. However, we do not agree with the reviewer for the point mentioned above. In our opinion, we discussed the AOM presence based on the bulk parameter at this part of the manuscript. So the point can be mentioned as discussed in our opinion.

L291. The authors first of all need to show methane data

Reply: As mentioned above, the methane data are not available for this study.

L304. How do the authors know these are sulfate-dependent AOM. If the abundance of sulfate-dependent AOM is elevated, it should be placed in the main text.

Reply: As suggested by the editor, we will move Fig. S1 and Table S1 into the main text in the revised version, but Fig. 2 into the Supplemenatry information as Fig. S1. Although we already mentioned the steep sulfate depletion related to AOM processes in the shallow depths (line 339-342) in the submitted version, we will extend the discussion on the sulfate methane transition zone in the revised version. We will mention that other electron acceptors such as nitrate/nitrite and oxidised Fe/Mn are typically not available at this depth, because the penetration depth of those ones is thermodynamically very limited.

L291-305. These sentences are mostly pointless. The evidence of AOM IN THIS STUDY should be first emphasized, and then discussed in the context of other studies.

Reply: As mentioned above, we will first emphasize evidence for sulfate-dependent AOM, which is typically the dominant methane oxidation process in marine sediment. On the other hand, AOM with alternative electron acceptors (e.g. Mn and Fe oxides)

in marine settings is probably mediated by specialised ANMEs, but it remains unclear how far potential partner bacteria are involved in these processes. Accordingly, we will address this point in the revised version.

L308-309. This can be placed in the introduction section.

Reply: We will revise it as recommended by the editor.

L311-312. To what extent, the author are certain that these DGD can represent sulfate-reducing bacteria? L313-314. If it is not supportive of SRB, it may suggest that other electronic acceptor such as Mn/iron/nitrate might be involved? Whatever, it cannot be stopped here and further discussion should be made.

Reply: This compound was previously identified as a diagnostic marker molecule for a sulfate-reducing bacterium in sediment where AOM was an important microbial process (Pancost et al., 2001). Moreover, we not only detected this compound but also measured its stable carbon isotopic composition, suggesting some degree of methane and other organic sources. As suggested by the editor, we will further explain the SRB-related DGD in the revised version.

L319. Do the author mean the contribution to GDGD, and so what?

Reply: We will discuss in more detail the contribution of archaea to the GDGT pool in the revised version. For example, we will incorporate a new aspect relating the relatively low $\delta$13C values of archaeol and biphytanes-derived from GDGTs with the activity of anaerobic methanotrophs. We will also address the potential carbon assimilation (e.g. chemoorganoautotrophs) of AOM-related archaea along the sharply depleted sulfate profiles.

L327-328. There is no solid evidence in support of this statement.

Reply: We will moderate our statement in the revised version and will emphasize the steeply depleted-sulfate profiles to strengthen our discussion with respect to the near-sedimentsurface S-cycle in the revised version.

L327-338. Much of these discussion appears more appropriate as the results

Reply: In our opinion, the significant AOM evidence could be shown from the detected biomarkers with a 13C depleted signature as the results of methane assimilation. As suggested by the editor, we do think that this paragraph can go into the results and then only the highlight should be kept here.

L339-341. What is the logic between the sulfate profiles and siboglinid tubeworms??? L344. What does the constrained mean?

Reply: This section aims at describing the indirect AOM influence on the surrounding ecosystem. For example that sulfide, an end product of sulfate dependent AOM can be utilized by thiotrophs such as symbiotic megafauna and free-living bacterial mats. We will strengthen this discussion in the revised version

L344-350. Maybe the author want to emphasize how sulfate is generated, and then used in support of methane oxidation. The paragraph needs to be re-organized.

Reply: We agree with the editor for this point. We will revised this part of the discussion in the revised version, as recommended by the editor.

L354-356. This should be placed in the introduction

Reply: We will revise it as recommended by the editor .

L354-375. Please clearly specify the lipids that are representative of different archaea. L380. What is ANME2-specific lipid?

Reply: As mentioned above, archaeol, sn-2 and -3-hydroxyarchaeol and to a lesser degree GDGTs are synthesized in diagnostic ratios by the different ANME groups. We will add a paragraph highlighting this and will then relate our findings to the previously detected ratios. In brief, Niemann and Elvert (2008) found that a sn2-archaeol:archaeol ratio of >1 is typical for ANME 2 archaea. We found a ratio of 1.3 to 1.8 and indeed, our 16S rRNA analyses showed an abundance of ANME2c and ANME3 in respect to

the corresponding values.

L407. To clarify

Reply: We will correct it.

L407-412. It can be described in the materials and method section

Reply: The mentioned chemotaxanomy/lipid data were used to identify different ANME groups. Hence, we would prefer to leave this as part of the discussion.

L412-424. These sentences are rather descriptive, and it might be more appropriate in the section of Results.

Reply: While the overall archaeal communities is indeed described in the results, we would prefer to leave this part on the methanomicrobia clade in the discussion. Particularly, the methanomicrobia OTUs c116, C1698 and C1784 forme a cluster with ANME-2c and -3. Without an introduction to the phylogeny, this part of the discussion would become tedious for the non-specialist in methanotrophic diversity. Thus, we also think that it is a suitable arrangement in this part of the discussion.

L430-437. What is your conclusion about AOM in this study, when compared to other studies?

Reply: Although we currently think that the distribution of ANME-2c and ANME-3 is probably controlled by the methane flux, it might be possible that other geochemical factors (e.g. oxgen and sulfide) influence these groups too. We will address this point in the revised version.

L437. Does this mean that in this study methane concentration is low?

Reply: Often (though not neccesarily), high fluxes are related to high pore fluid mathen cocnetrations. However, when comparing the different ANME2 subclusters (a versus c) then ANME2c appears to prefer niches with lower methane fluxes

L444 and L450. Where is the measured data of methane flux

Reply: We cannot present flux data becuase we have not CH4 concentraton data. However the preferences to differential flux regimes shown in the literature indicate differential flux regimes. We inferred differences in ANME groups through the thermal gradients (calculated from heat flow in core sediments) as mentioned above, although methane data are not available for this study.

L454-464. Part of this discussion should be made in the result section.

Reply: For this point, we do not agree with the editor. In our opinion, the distribution of confirmed ANMEs was closely related to methane flux, like the Haakon Mosby Mud Volcano located in the Barents Sea. Thus, this discussion is important to identify different distrubitions of ANMEs along with the variation of methane flux.

L476-477. The authors need to specify that these sn-2- and sn-3-hydroxyarchaeol are representative of AOM at the very beginning.

Reply: We will revise it as recommended by the editor.

L480. There is no evidence of methane concentration. How could the author claim that methane was oxidized?

Reply: Our lipid and microbial data clearly showed the presence of AOM communities in our study sites. Most importantly, the depleted ïĄď13C values show clear signs of methane derived carbon incorporation, suggesting an active ANME-2 and -3 community.

As for the Tables and Figures, they need to be significantly re-structured.

Reply: We revised the tables and figures as recommended by the editor.

Fig. 1 should be re-organized. Fig. 1B could be placed at the bottom left. Fig. C could be place at top right. Fig. 1D appears to be the most important one which could have more space like the current Fig. 1B, and place in the middle right.

Reply: We revised it as suggested.

Fig. 2. In a scientific paper it is unusual to show an example figure. This figure tells the readers very little information, and it should be placed in the supplementary, or it should be placed side by side with the data measurements.

Reply: We replaced it in the supplementary.

The title of this ms is about biogeochemical. Therefore, in the main text, the BIO and the Geochemical data should be included. But all figures are about the BIO evidence.

Reply: We revised it as suggested.

Table S1 should be placed in the main text as the figure 2. Methane concentration is of particular concern, and should be placed together with sulfate gradient.

Reply: We revised it as suggested.

The most important data that are related to AOM in Table S1 should be made as a figure and placed in the main text.

Reply: The figures related Table S1 are illustrated as Fig. 3 and 4.

What are the key information of Fig. 3 and Fig .4. these data appear to be from Table S1. Please stay focused on the AOM as much as possible.

Reply: In our opinion, it is good to show the data as figures to see the pattern although the exact data are presented as a table.

Please also note the supplement to this comment:
https://www.biogeosciences-discuss.net/bg-2018-91/bg-2018-91-AC2-supplement.pdf
* * *
[Figure]

A)
Beaufort Sea
Beaufort Shelf
CANADA
Alaska
Canada
50 km
140°W
130°W
70°N
B)
C)
MV420 (ARA05C-01-GC)
MV740 (ARA05C-18-GC)
D)
MV282
(ARA05C-10-GC)
MV420
(ARA05C-01-GC)
MV740
(ARA05C-18-GC)
Core depth (mbsf)
0
1
2
3
Total length
221 cm
Total length
272 cm
Total length
300 cm
LEGEND
Studied mud volcanoes
Sampling sites
Brown silty-clay
Silty-clay
Mousse-like texture
Void layers
Gas hydrates
0
5
10
20
Kilometers
-800
-600
-400
-200
MV740
MV420
MV282
70°50'0"N
70°40'0"N
136°00'0"W
135°30'0"W

**Fig. 1.**

[Figure]

**Fig. 2.**

[Figure]

Lipid biomarker concentrations
Archaeol
sn-2-hydroxyarchaeol
sn-3-hydroxyarchaeol
DGD (lf)
A)
B)
C)
Archaeol (µg/ g dw)
Hydroxyarchaeols (µg/ g dw)
DGD (lf) (µg/ g dw)
sn-2-hydrxyarchaeol / archaeol
Core depth (mbsf)
MV282 (ARA05C-10-GC)
MV420 (ARA05C-01-GC)
MV740 (ARA05C-18-GC)
Lipid biomarker δ13C values
Archaeol
sn-2-hydroxyarchaeol
sn-3-hydroxyarchaeol
DGD (lf)

Fig. 3.

[Figure]

A) MV282 (ARA05C-10-GC)
B) MV420 (ARA05C-01-GC)
C) MV740 (ARA05C-18-GC)
Core depth (mbsf)
0
1
2
3
I
II
III
IV
V
Relative intensity
Retention time

Fig. 4.

[Figure]

**Fig. 5.**

**Supplement:**

1 **Supplemetary information**

2 Table S1. Summary of pyrosequencing reads.

| Sites | Core depth (mbsf) | Summary of SSU rRNA tags | | | | | Diversity index | | |
|---|---|---|---|---|---|---|---|---|---|
| | | Number of total reads | Archaeal reads | Bacterial reads | Eukaryotic reads | Unknown reads | Shannon | Simson | Ace |
| MV282 | 0.02 | 12214 | 8835 | 134 | 2790 | 455 | 1.53 | 0.34 | 36.34 |
| | 0.09 | 8875 | 7016 | 36 | 1652 | 171 | 1.60 | 0.33 | 187.09 |
| | 0.20 | 8222 | 8060 | 24 | 53 | 85 | 1.56 | 0.28 | 33.15 |
| | 0.33 | 7304 | 7224 | 5 | 34 | 41 | 1.05 | 0.44 | 40.90 |
| | 0.50 | 6182 | 6157 | 9 | 9 | 7 | 1.11 | 0.42 | 144.99 |
| | 0.88 | 8886 | 8780 | 13 | 32 | 61 | 0.96 | 0.46 | 24.14 |
| | 1.05 | 6283 | 6266 | 0 | 15 | 2 | 1.03 | 0.45 | 24.93 |
| | 1.30 | 5058 | 5005 | 6 | 10 | 37 | 1.06 | 0.44 | 33.37 |
| | 1.60 | 1902 | 1875 | 2 | 9 | 16 | 1.33 | 0.38 | 34.30 |
| | 1.90 | 3550 | 3542 | 0 | 4 | 4 | 0.92 | 0.51 | 131.86 |
| MV420 | 0.08 | 3155 | 3028 | 23 | 29 | 75 | 1.28 | 0.55 | 104.50 |
| | 0.20 | 4189 | 4079 | 22 | 32 | 56 | 2.68 | 0.12 | 125.60 |
| | 0.33 | 5164 | 1436 | 64 | 3508 | 156 | 2.84 | 0.11 | 68.23 |
| | 0.50 | 2175 | 2041 | 1 | 44 | 89 | 2.49 | 0.15 | 133.00 |
| | 0.70 | 2307 | 2259 | 1 | 3 | 44 | 2.08 | 0.24 | 74.00 |
| | 1.05 | 1537 | 1520 | 0 | 10 | 7 | 1.47 | 0.44 | 102.82 |
| | 1.38 | 5207 | 4757 | 71 | 107 | 272 | 1.74 | 0.35 | 40.90 |
| | 1.60 | 7012 | 6985 | 6 | 8 | 13 | 1.45 | 0.42 | 37.70 |
| | 1.81 | 3706 | 3669 | 5 | 7 | 25 | 1.40 | 0.45 | 52.58 |
| | 2.17 | 12017 | 11865 | 11 | 80 | 61 | 1.36 | 0.47 | 75.61 |
| MV740 | 0.08 | 522 | 445 | 17 | 29 | 31 | 3.59 | 0.05 | 178.09 |
| | 0.20 | 506 | 458 | 0 | 28 | 20 | 3.04 | 0.11 | 170.50 |
| | 0.35 | 674 | 589 | 1 | 75 | 9 | 3.12 | 0.09 | 112.57 |
| | 0.45 | 583 | 534 | 0 | 35 | 14 | 2.84 | 0.10 | 82.31 |
| | 0.55 | 706 | 673 | 1 | 6 | 26 | 2.92 | 0.09 | 97.31 |

Table S2. Heat map and taxonomic affiliation three dominant methanomicrobial OTUs along the depth. The color gradient from white to brown indicates low to high relative abundance values.

| OTU_ID | Taxonomy | | | | | | MV282 Core depth (mbsf) | | | | | | | | | | MV420 Core depth (mbsf) | | | | | | | | | | MV740 Core depth (mbsf) | | | | |
|---|---|---|---|---|---|---|---|---|---|---|---|---|---|---|---|---|---|---|---|---|---|---|---|---|---|---|---|---|---|---|---|
| | phylum | class | order | family | genus | species | 0.02 | 0.09 | 0.20 | 0.33 | 0.50 | 0.88 | 1.05 | 1.30 | 1.60 | 1.90 | 0.08 | 0.20 | 0.33 | 0.50 | 0.70 | 1.05 | 1.38 | 1.60 | 1.81 | 2.17 | 0.08 | 0.20 | 0.35 | 0.45 | 0.55 |
| c116 | Euryarchaeota | Methanomicrobia | Methanosarcinales | ANME3_f | ANME3_g | ANME3_s | 2.5 | 2.0 | 14.1 | 0.0 | 0.0 | 0.0 | 0.0 | 0.0 | 0.0 | 0.1 | 5.5 | 3.4 | 6.1 | 0.2 | 6.7 | 0.5 | 0.0 | 0.0 | 0.0 | 0.0 | 0.0 | 0.2 | 0.0 | 0.0 | 0.0 |
| c1698 | Euryarchaeota | Methanomicrobia | Methanosarcinales | ANME3_f | ANME3_g | ANME3_s | 1.3 | 0.9 | 0.0 | 0.0 | 0.0 | 0.0 | 0.0 | 0.0 | 0.0 | 0.0 | 0.9 | 0.3 | 0.4 | 0.0 | 0.0 | 0.0 | 0.0 | 0.0 | 0.0 | 0.0 | 0.0 | 0.0 | 0.0 | 0.0 | 0.0 |
| c1784 | Euryarchaeota | Methanomicrobia | Methanosarcinales | ANME2_f | ANME2c_g | DQ084449_s | 1.7 | 3.4 | 6.8 | 1.2 | 0.0 | 0.0 | 0.1 | 0.0 | 0.0 | 0.7 | 0.0 | 0.0 | 0.1 | 0.9 | 1.3 | 0.3 | 0.0 | 0.0 | 0.1 | 0.1 | 0.0 | 0.7 | 3.7 | 11.2 | 14.9 |

10    Fig. S1. Examples of GC-MS chromatograms of polar fractions obtained from sediment cores

11    (A) ARA05C-10-GC (MV282): core depth 0.1 m, (B) ARA05C-01-GC (MV420): core depth

12    0.7 m, and (C) ARA05C-18-GC (MV740): core depth 0.8 m. Solid triangles denote *n*-

13    alcohols.

[Figure]

14

15  Fig. S2. Relative abundances of archaeal communities at the class level along the depth (A)

16  MV282, (B) MV420 and (C) MV740.

[Figure]

17

---

## Author Response (AR1)

Korea Polar Research Institute

**Jung-Hyun Kim**
Korea Polar Research Institute (KOPRI),
26 Songdomirae-ro, Yeonsu-gu, Incheon 21990, South Korea
Telephone: (+82) (0)32-7605377
Fax: (+82) (0)32-7605377
E-mail: jhkim123@kopri.re.kr

Submission of the revised article "**Biogeochemical evidence for anaerobic methane oxidation at active submarine mud volcanoes on the Canadian Beaufort Sea slope**" by Lee et al.

Incheon, October 30 2018

Dear associate editor Zhongjun Jia,

We sincerely thank you for providing us an opportunity to revise this article (ID: bg-2018-91). Please find the revised version of the article. We revised the text as you suggested, after uploading the revised tables and figures in the journal website as required before. The references were also up-dated in the revised version. The changed parts of the text were highlighted by blue color.

We hope our article will be considered positively for publication in Biogeosciences.

Yours sincerely,

Jung-Hyun Kim

---

## Author Response (AR2)

**Associated editor**
Manuscript ID: bg-2018-91
Biogeochemical evidence of anaerobic methane oxidation on active submarine mud volcanoes on the continental slope of the Canadian Beaufort Sea

The key concern I would like to point out is that the writing of abstract. Please remove the statements that might be controversial, and that are not widely accepted. For example, phylogenetic analysis of 16S rRNA gene could be considered as the ultimate evidence for the presence of ANME, but the ratio of of sn-2-hydroxyarchaeol to archaeol (>1) might not be that conclusive. In addition, the general term ANME-specific lipid could be used rather than sn-2-hydroxyarchaeol in the abstract, unless these lipids are solely detected in ANME. In the introduction and discussion, the authors can then discuss the advantage and disadvantages of these lipids as biomarkers of ANMEs. In a word, one has to bear it in mind that sequencing analysis of 16S rRNA genes is more conclusive evidence than the lipid-based chemotaxonomy.
Reply: We thank the associated editor for the constructive comments. We revised the abstract according to the comments made by the associate editor.

Minor comments
(1) L34-36. How confident the authors are by pointing out that these sn-2 and sn-3 hydroxyarchaeol are ANME-specific biomarker. If it was also detected in some of methanogens, the authors might have to rephrase these sentences.
Reply: We clarified this sentence as follows: The carbon isotopic compositions ($\delta^{13}$C) of sn-2- and sn-3-hydroxyarchaeol showed the highly $^{13}$C-depleted values (–114 ‰ to –82 ‰) associated with a steep depletion in sulfate concentrations within 0.7 m of sediment depths. This suggested the presence of methanotrophic archaea involved in sulfate dependent–AOM, albeit in a small amount.

(2) L36-37. I guess phylogentic analysis alone is sufficient to claim that ANME-2c and 3 are the predominant methane oxidizers. The ratio of sn-2-hydroxyarchaeol to archaeol may not be that conclusive.
Reply: Although the ratio of sn-2-hydroxyarchaeol to archaeol is not that conclusive compared to the phylogenic analyses, it can still provide the first clue for their presence.

(3) L42. Please conclude the abstract by stating the importance and/or implication of this study, but not the perspective for future study. For example, these results suggest that niche diversification of active mud volcanoes has shaped distinct archaeal communities that play important roles in anaerobic methane oxidation and organic matter turnover in Beaufort Sea.
Reply: We revised this sentence as suggested: Consequently, our results suggest that the niche diversification of active mud volcanoes has shaped distinct archaeal communities that play important roles in AOM in the Beaufort Sea.

(4) The redundant description? i.e., The content of L53-56 appears to be similar to that of Line 60-63.
Reply: We agree with the associated editor. Thus, we deleted the content of L53-56.

(5) L66-67. There is no necessary to cite 6 different references for a simple statement. Please delete some

Reply: we deleted some references (i.e. Steele et al., 2008; Thatcher et al., 2013, Somavilla 2013).

(6) L107. by using a combination suite of lipid and nucleic acid analyses...
Reply: We revised this sentence as suggested.

(7) L213. Please specify the samples used for DNA extraction
Reply: We specified it as suggested.

(8) L257, Please indicate that these archaeol can be considered as the biomarker for ANME.
Reply: We clarified specific biomarkers (i.e. archaeol and sn-2-hyroxyarchaeol).

(9) L281. Rephrase as "Depth profile of archaeal communities"
Reply: We revised it.

(10) L282. It can be rephrased as following: Archaeal communities were phylogenetically classified as the taxonomic level of class
Reply: We revised it as recommended.

(11) L326. Do the authors determine the concentration of others electronic acceptors such as Mn and Nitrate and Fe?
Reply: Unfortunately, other electronic acceptors are not available for this study. However, we discussed potential possibilities for AOM coupled with other electron acceptors in the previously submitted version. Consequently, by using a combined suite of lipid and nucleic acid analyses, we could provide the evidence of sulfate-dependent AOM in our study sites.

(12) L331. One cannot get a clear idea of this section,. i.e., it the AOB biomass contribution to TOC is low, so what?
Reply: In the case of mud volcanoes where methane is actively venting as free gases, methanotrophs are not easy to assimilate methane as their carbon sources. Indeed, with respect to narrow ranges of $\delta^{13}C_{TOC}$ values, we infer that the biomass of methanotrophic communities inhabiting the Beaufort mud volcanoes comprises a minor portion in the total organic carbon pool. Thus, this seems to be linked to the relatively low abundances of AOM biomass in accordance with active methane fluxes in the Beaufort mud volcanoes. Nevertheless, by using a suite of lipid and nucleic acid analyses, our results suggest that distinct archaeal communities (ANME-2c and ANME-3) inhabit the Beaufort mud volcanoes playing an important role in AOM in the Beaufort Sea, albeit in a small amount.

(13) L347. In the result section, please briefly mention that these non-isoprenoid DGD could be considered as a potential marker of sulfate-reducing bacteria
Reply: Based on previous studies, we already mentioned that in the submitted version previously. But, we clarified this sentence in the revised version

(14) L385. Delete and refernices therein. It has been mentioned
Reply: We deleted it.

(15) L391-396. It seems contradictory to some extent. For example, In the abstract, the authors claimed that both sn-2-hydroxyarchaeol and sn-3-hydroxyarchaeol are representative of ANMEs, But in this paragraph, it appears that these archaeol could be found also in methanogens. The question is how specific these biomarkers for ANMEs

Reply: The methanotrophic evidence inferred from these biomarkers can be obtained from the $^{13}$C-depleted values as the result of their carbon assimilations by mentioned organisms. Actually, previous studies reported the predominance of methanotrophs which is related to the methanogenic orders *Methanomicrobiales* and *Methanosarcinales* (Hinrich et al., 1999). Thus, considering $\delta^{13}$C values of methane in our study sites, the occurrence of $^{13}$C-depleted lipids suggests the presence of methanotrophic archaea rather than methanogens. This highlights the importance of the combination of biomarkers works with the isotopic analyses.

(16) L399. Does this mean that sn-2-hydroxyarchaeol is a better indicator for ANME-2, while archaeol can represent ANME-1 much better?

Reply: In accordance with the currently available literature data, each archaeal biomarker showed differences in abundance among methanotrophs (ANME-1, -2 and -3) (Niemann and Elvert, 2008). In this regard, the ratio of these compounds was proposed as a specific fingerprint for different ANMEs. Particularly, $^{13}$C-depleted hydroxyarhcaeol have been regarded as specific ANME markers (mostly ANME-2). Consequently, together with archaeal taxanomic results, we discussed different distributions of ANMEs inhabiting each mud volcano where fluid fluxes were varied.

(17) L453. MCG group now refers to Bathyarchaeota ?
Reply: We revised it.

(18) L466. at a site?
Reply: We corrected it.

(19) Fig.1. please provide the full name of mbsf
Reply: We added the information.

(20) Fig.5. if flexible, please add the bootstrap value of >70 on the tree
Reply: We revised it.